# Converging synaptic and network dysfunctions in distinct autoimmune encephalitis

Daniel Hunter [1,5], Mar Petit-Pedrol [1,5], Dominique Fernandes [1], Nathan Bénac[1], Catarina Rodrigues [1], Jakob Kreye[2,3], Mihai Ceanga[4], Harald Prüss [2,3], Christian Geis[4] & Laurent Groc [1✉]

## Abstract

**Psychiatric and neurological symptoms, as well as cognitive deficits, represent a prominent phenotype associated with variable forms of autoimmune encephalitis, regardless of the neurotransmitter receptor targeted by autoantibodies. The mechanistic underpinnings of these shared major neuropsychiatric symptoms remain however unclear. Here, we investigate the impacts of patient-derived monoclonal autoantibodies against the glutamatergic NMDAR (NMDAR mAb) and inhibitory GABAaR (GABAaR mAb) signalling in the hippocampal network. Unexpectedly, both excitatory and inhibitory synaptic receptor membrane dynamics, content and transmissions are altered by NMDAR or GABAaR mAb, irrespective of the affinity or antagonistic effect of the autoantibodies. The effect of NMDAR mAb on inhibitory synapses and GABAaR mAb on excitatory synapses requires neuronal activity and involves protein kinase signalling. At the cell level, both autoantibodies increase the excitation/inhibition balance of principal cell inputs. Furthermore, NMDAR or GABAaR mAb leads to hyperactivation of hippocampal networks through distinct alterations of principal cell and interneuron properties. Thus, autoantibodies targeting excitatory NMDAR or inhibitory GABAaR trigger convergent network dysfunctions through a combination of shared and distinct mechanisms.**

**Keywords** Autoantibody; Encephalitis; Excitation; Inhibition; Neurology
**Subject Categories** Immunology; Molecular Biology of Disease; Neuroscience

## Introduction

Recent years have seen the identification of an array of autoimmune neuropsychiatric disorders with patients expressing autoantibodies directed against membrane proteins. Two prominent diseases in this category are anti-NMDAR encephalitis and anti-GABAaR encephalitis, in which patients develop antibodies directed against major excitatory or inhibitory ionotropic channels, respectively (Crisp et al, 2016; Dalmau et al, 2007; Petit-Pedrol et al, 2014). Most of the molecular investigations have delineated an acute pathway of action of these autoantibodies, focusing on the isolated impacts on the target antigens (Carceles-Cordon et al, 2020). Anti-NMDAR autoantibodies (NMDAR-Abs) bind to the extracellular N-terminal domain of the obligatory GluN1 subunit (Gleichman et al, 2012; Hughes et al, 2010). Antibody binding causes a synaptic destabilisation of the NMDARs through altered interaction with the EphB2 receptor (Mikasova et al, 2012; Planaguma et al, 2016). This destabilisation causes an increase in the surface dynamics of synaptic NMDARs at acute timescales, promoting displacement to extrasynaptic sites. The overall amount of cell-surface NMDARs decreases, possibly through a promoted internalisation, resulting in an excitatory hypofunction on hippocampal neurons (Hughes et al, 2010; Ladepeche et al, 2018; Mikasova et al, 2012). For anti-GABAaR autoantibodies (GABAaR-Abs), some can elicit a direct antagonism of the inhibitory ion channel whereas others are without antagonistic action (Kreye et al, 2021; Noviello et al, 2022; van Casteren et al, 2022). Direct antagonism of GABAaR channels has been demonstrated to occur through competition by GABAaR-Abs for GABA binding sites at the interface between α and β subunits, and also through negative allosteric modulation of the channel (Noviello et al, 2022). Such antagonism drives a hypofunction of the major inhibitory neurotransmission system. The effect of GABAaR-Abs on the receptor surface organisation and diffusion has however not been examined. Thus, NMDAR-Abs and GABAaR-Abs drive either excitatory or inhibitory hypofunction, respectively, in hippocampal networks.

Despite the obvious opposition in the function of these autoantibody targets, seizure phenotypes and cognitive deficits form a prominent facet of shared symptomology in these two autoimmune diseases (Dalmau et al, 2019; Geis et al, 2019; Kreye et al, 2021; Petit-Pedrol et al, 2014; Titulaer and Dalmau, 2014; Wright and Vincent, 2016; Wright et al, 2021). In the case of GABAaR encephalitis, antagonism of inhibitory GABAergic

[1]University of Bordeaux, CNRS, Interdisciplinary Institute for Neuroscience, IINS, UMR 5297, F-33000 Bordeaux, France. [2]German Center for Neurodegenerative Diseases (DZNE) Berlin, 10117 Berlin, Germany. [3]Department of Neurology and Experimental Neurology, Charité-Universitätsmedizin Berlin, Corporate Member of Freie Universität Berlin, Humboldt-Universität Berlin, 10117 Berlin, Germany. [4]Hans-Berger Department of Neurology, Jena University Hospital, Jena, Germany. [5]These authors contributed equally: Daniel Hunter, Mar Petit-Pedrol. ✉E-mail: laurent.groc@u-bordeaux.fr

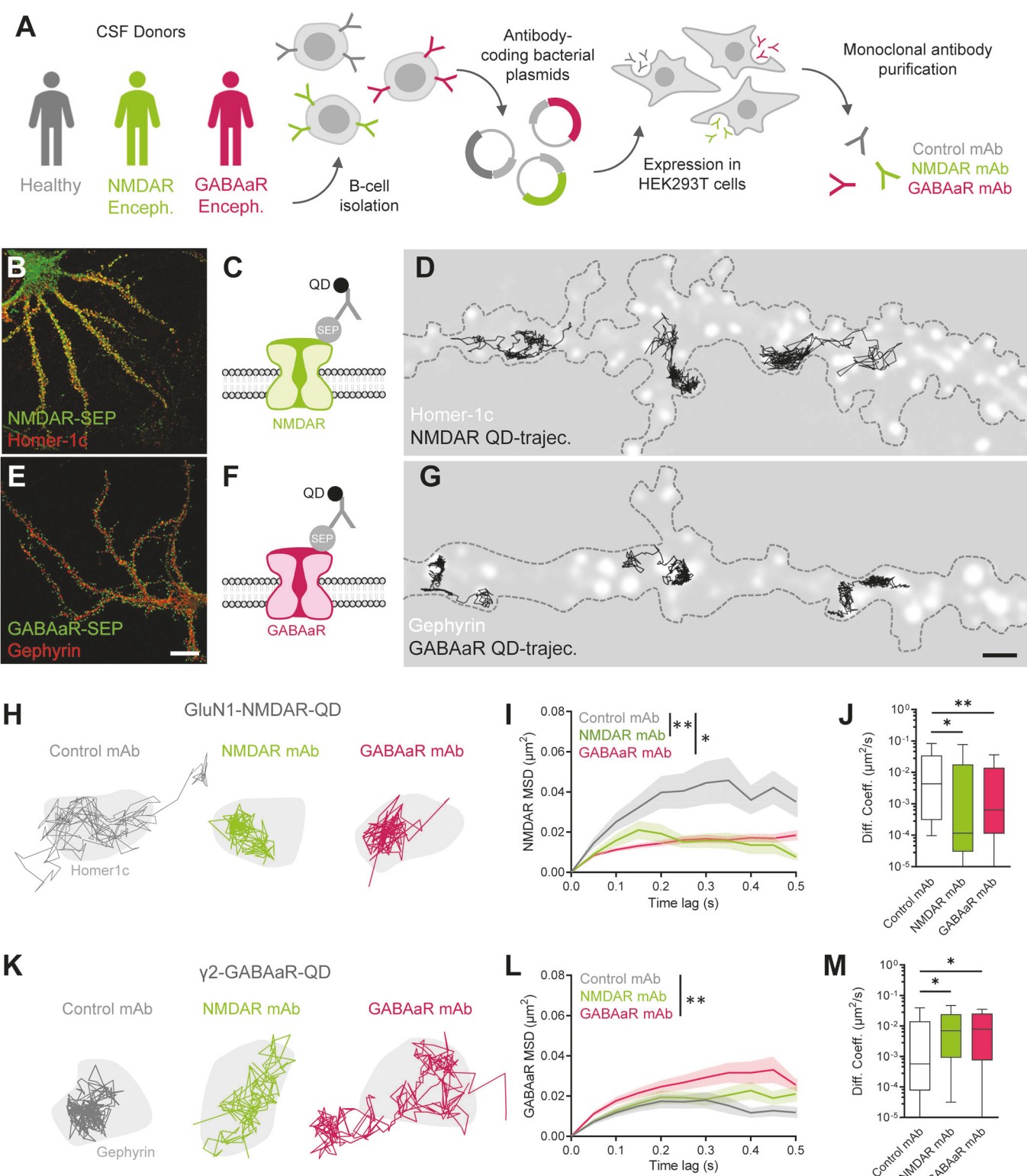

transmission would intuitively lead to epileptogenesis, where the loss of inhibitory control results in over-excitation in brain regions that are dense in recurrent microcircuits such as the temporal lobe. It is not surprising then that ~80% of patients in this cohort present with severe seizure phenotypes, elevating this to the predominant

disease feature of GABAaR encephalitis (Dalmau et al, 2019; Geis et al, 2019; Kreye et al, 2021; Petit-Pedrol et al, 2014; Titulaer and Dalmau, 2014). The mechanisms by which NMDAR-Abs induce excitatory hypofunction at the molecular scale result in a similar phenotype remains enigmatic. Yet, seizures are observed in ~80%

**Figure 1. Patient autoantibodies mutually disrupt the surface dynamics of NMDA receptors and GABAa receptors.**

(A) Schematic representation of experimental protocols for the derivation and production of human monoclonal autoantibodies. (B) Representative image of cultured hippocampal neuron expressing GluN1-SEP (green) and Homer1c-dsRed (red). Scale bar = 20 μm. (C) Schematic representation of quantum dot (QD) tracking method of NMDAR. (D) Illustrative schematic to show the tracking of NMDAR (solid black line) in the membrane of the neuronal dendrite (dashed line). Scale bar = 1 μm. (E) Representative image of cultured hippocampal neuron expressing γ2-SEP (green) and gephyrin-mVenus (red). Scale bar = 20 μm. (F) Schematic representation of quantum dot (QD) tracking method of GABAaR. (G) Illustrative schematic to show the tracking of GABAaR (solid black line) in the membrane of the neuronal dendrite (dashed line). Scale bar = 1 μm. (H) Representative single molecule tracks of NMDAR in synaptic compartments, after 24 h exposure to Control (grey), NMDAR (green) or GABAaR mAb (magenta). (I, J) Mean square displacement and diffusion coefficients of synaptic NMDAR trajectories (Control mAb $n = 66$ trajectories; NMDAR mAb $n = 80$; GABAaR mAb $n = 109$; Kruskal–Wallis test), error bar areas represent the standard deviation. Box plots represent the median and interquartile range (25–75%), with the min and max values. (K) Representative single molecule tracks of GABAaR in synaptic compartments, after 24 h exposure to Control mAb (grey), NMDAR mAb (green) or GABAaR mAb (magenta). (L, M) Mean square displacement and (M) diffusion coefficients of synaptic GABAaR trajectories (Control mAb $n = 66$ trajectories; NMDAR mAb $n = 61$; GABAaR mAb $n = 89$; Kruskal–Wallis test). Box plots represent the median and interquartile range (25–75%), with the min and max values. Data information: All error bars represent the standard error of the mean. Significance levels are represented as *$P < 0.05$, and **$P < 0.01$.

of cases, forming a considerable segment of clinical presentation along with psychiatric dysfunction (90%) and movement disorder (78%) (Dalmau et al, 2019). It has been suggested that NMDAR-Abs may display a degree of selectivity in pathogenesis whereby inhibitory interneuron populations are primarily impaired (Hunter et al, 2021). Some experimental evidence has begun to emerge in support of this hypothesis, suggesting a disinhibitory action of NMDAR-Abs in cortical networks that lead to hyperexcitation and network hypersynchrony based on disrupted excitatory-inhibitory balance and neuronal excitability (Andrzejak et al, 2022; Ceanga et al, 2023; Wright et al, 2021). These findings are particularly intriguing considering the proposed role of interneuron-specific hypofunction in other psychiatric conditions such as schizophrenia and bipolar disorder, which bare resemblance to early stages of NMDAR encephalitis (Guasp et al, 2022; Marin, 2016; Muniz-Castrillo et al, 2022). Seizure represents the clinical endpoint of significant perturbation to excitatory-inhibitory (E/I) balance and network homeostasis. The maintenance of physiological E/I balance in the healthy brain is supported by a complex interplay between many cell-autonomous and network-level systems, including modulations of synaptic plasticity, scaling, synaptic density and intrinsic cellular excitability (Chen et al, 2022; Kullander and Topolnik, 2021). Excitatory and inhibitory systems exert mutual influence over each other in order to reach a physiological equilibrium. This can be achieved through intracellular signalling cascades, for example, the bidirectional linking of NMDAR-mediated calcium influx to local GABAaR mobility and reorganisation (Bannai et al, 2015). In addition, common interacting proteins such as type I and II dopamine receptors, which bind NMDA- and GABAa receptors, respectively, can act as centralised orchestrators of excitatory and inhibitory signalling (Ladepeche et al, 2014; Lee et al, 2002; Liu et al, 2000; Maingret and Groc, 2021; Schoffelmeer et al, 2000).

Given the array of homeostatic systems in place, each dependent on effective crosstalk between excitatory and inhibitory signalling, the mechanistic underpinnings of autoimmune encephalitis may reside in the pathological co-regulation of excitatory and inhibitory receptors and transmissions. Yet, the characterisation of this cross-synaptic disruption remains to be determined. Here, we intended to tackle the issue on how autoantibodies that induce glutamatergic hypofunction or GABAergic hypofunction produce a similar set of symptoms in patients and behavioural deficits in rodents. Specifically, we investigated how patient-derived monoclonal autoantibodies against the excitatory NMDAR (NMDAR mAb) and inhibitory GABAaR (GABAaR mAb) (Fig. 1A) impact

excitatory and inhibitory drive and cell properties in the hippocampus, spanning from single molecule to neuronal network exploration.

## Results

### Patient autoantibodies similarly alter NMDAR and GABAaR membrane diffusion

To shed light on the effect of autoantibodies on their targets, we performed single Quantum Dot (QD) tracking experiment to define the membrane diffusion of GluN1-containing NMDARs and γ2 subunit containing GABAaRs in cultured hippocampal neurons. Synaptic areas were defined by expressing homer1c-dsRed to locate glutamatergic synapses or gephyrin-mCherry to locate GABAergic synapses (Fig. 1B–G). We then exposed neuronal networks to control, NMDAR or GABAaR mAb for 24 h before tracking membrane NMDAR and GABAaR. We selected the NMDAR mAb clone 003–102 and the GABAaR mAb clone 113–115 that partially inhibits GABAaR-mediated current (Kreye et al, 2021). These high-affinity autoantibodies were used at the dose of 0.5–1 μg/ml since their concentrations in patients with encephalitis has been estimated at 0.1–5 μg/ml and their maximal binding capacity onto neurons around few μg/ml (Ly et al, 2018). The NMDAR mAb treatment did not alter the glial and microglia coverage in the hippocampus, as measured by the immunostaining of GFAP- and Iba1-positive cells (Fig. EV1). As expected, NMDAR mAb alter the membrane diffusion of synaptic NMDARs (Fig. 1H) by inducing a decrease in the mean square displacement (MSD) curve, i.e., trajectories are more confined, and a significant decrease in the diffusion coefficient (Fig. 1I-J). Surprisingly, analysis of GABAaR surface diffusion also revealed that NMDAR mAb significantly alter GABAaR surface trafficking at inhibitory synapses (Fig. 1k). In this case, we observed an overall increase in the surface diffusion of GABAaRs, reflected by a left shift in the MSD curve, i.e., trajectories are less confined, and a significant increase in the diffusion coefficient (Fig. 1L,M). This indicates a cross-synaptic alteration in the trafficking of both NMDARs and GABAaRs in the context of anti-NMDAR autoimmune encephalitis. After exposure of neuronal networks to GABAaR mAb, surface GABAaR were more diffusive within inhibitory synaptic compartments (Fig. 1K). Specifically, GABAaR mAb induced a left shift in the MSD curve (Fig. 1L,M). In addition, synaptic NMDAR diffusion was also altered by GABAaR mAb (Fig. 1H), as evidenced by inducing a

right shift in the MSD curve, i.e., trajectories are more confined, and a significant decrease in the diffusion coefficient (Fig. 1I,J). It is striking to note that, not only do both NMDAR and GABAaR mAb elicit cross-synaptic deficits, but the resulting perturbations in channel dynamics essentially mimic each other. Altogether, these data indicate that NMDAR- and GABAaR mAb alter the membrane diffusion of their target as well as another key receptor, suggesting a major change in both excitatory and inhibitory synapses.

## NMDAR and GABAaR mAb similarly reduce excitatory synaptic content

To characterise the impacts elicited by NMDAR and GABAaR mAb at a macroscopic level, we examined the receptor content at glutamatergic synapses by quantifying the expression and localisation of two glutamatergic receptors: NMDAR and GluA1-containing AMPAR. To achieve staining of surface proteins neurons were transfected with receptor subunits tagged at their extracellular N-termini. Hippocampal neurons were exposed to control, NMDAR or GABAaR mAb for 24 h prior to imaging at 14 days in vitro (DIV). Given the above mutual defect of NMDAR membrane diffusion induced by both pathogenic autoantibodies, one may predict a similar alteration of the receptor synaptic content. Exposure to both NMDAR and GABAaR mAb elicited a significant reduction in NMDAR cluster density and area (Fig. 2A–C), suggesting that chronic exposure to both patient autoantibodies induce a hypo-content of synaptic NMDARs. Similarly, NMDAR and GABAaR mAb induce a moderate but significant reduction in the area of synaptic AMPAR clusters, although they did not impact the cluster density (Fig. 2D–F). Given that NMDAR and AMPAR are stabilised at synapses, at least in part, by association with scaffolding proteins that are embedded within the postsynaptic density, we assessed the possibility that the postsynaptic density is impacted after 24 h of autoantibody exposure. Consistently, Homer1c cluster density and area were slightly decreased in the presence of autoantibodies (Fig. 2I–H). Because encephalitis patients likely express more than one clone of autoantibodies, we tested the impact of two different autoantibodies: NMDAR mAb clone 008–218 (lower affinity than the above clone 003–102) and GABAaR mAb clone 113–175 (reduced affinity and no antagonistic effect on GABAaR-mediated current) (Kreye et al, 2021; Ly et al, 2018). The NMDAR mAb clone 008–218 decreased NMDAR, as well as GABAaR, cluster area and density to the same extent as clone 003–102 (Fig. EV2A–C). The GABAaR mAb clone 113–175 decreased synaptic GABAaR, as well as NMDAR, cluster density and area to the same extent as clone 113–115 (Fig. EV2D–F). The AMPAR and Homer1c clusters were also decreased by both mAbs (Fig. EV2G,H). Thus, different mAbs produced the same cross-effect on glutamatergic and GABAergic receptors, irrespective to their affinities (NMDAR) or antagonistic effect on the ionotropic current of the target receptor (GABAaR). Based on these findings, we then tested whether mAb-induced alterations in glutamatergic synaptic content is mimicked by reduced transmission of the NMDAR. To explore this, we exposed neuronal cultures to a partially blocking concentration (1 μM) of the NMDAR antagonist APV for 24 h. This concentration was selected as to mimic the incomplete reduction of NMDAR signalling induced by NMDAR mAb exposure (Hughes et al,

2010; Mikasova et al, 2012; Wright et al, 2021). This pharmacological manipulation failed to recapitulate any alteration in NMDAR or AMPAR localisation to glutamatergic synapses (Fig. 2G), suggesting that the autoantibody effects are primarily triggered by alteration in the surface dynamics of membrane proteins.

## NMDAR and GABAaR mAb similarly disrupt inhibitory synaptic content and scaffold proteins

Having uncovered substantial dysregulation at excitatory glutamatergic synapses by both encephalitis-derived autoantibodies, we further investigated the content and structuration of inhibitory synaptic compartments. Analysis of GABAaR content at inhibitory synapses demonstrated that both NMDAR- and GABAaR mAb decreased the synaptic content of GABAaRs. We report a significant reduction in synaptically localised GABAaR cluster density, with no alteration in cluster area (Fig. 2J–L). Given that synaptic GABAaR diffusion is regulated in part by scaffolding proteins (Battaglia et al, 2018), we investigated the potential for autoantibody-mediated disturbance of the inhibitory synaptic scaffolding protein, gephyrin. In neurons exposed to either NMDAR or GABAaR mAb for 24 h, we observed a clear reduction in the density of gephyrin clusters along dendritic segments (Fig. 2M,N) without affecting gephyrin cluster area (Fig. EV2I). Since gephyrin phosphorylation can tune its synaptic clustering (Tyagarajan et al, 2011; Zacchi et al, 2014), we explored the phosphorylation status of gephyrin. Specifically, phosphorylation of a serine residue at position 270 (S270) has been demonstrated to induce a reduction in gephyrin cluster densities, through the enhanced recruitment and activation of calpain, a protease that subsequently cleaves the scaffolding protein (Tyagarajan et al, 2011). We exposed neurons to control, NMDAR or GABAaR mAb for 24 h prior to immuno-labelling for total gephyrin expression or with a phosphorylation-specific antibody, which binds selectively to gephyrin scaffolds that have been phosphorylated at S270 (Fig. 3D) (Kuhse et al, 2012). As expected in control condition, the density of phosphorylated gephyrin clusters was significantly lower compared to total gephyrin (Fig. 2O). However, phosphorylated (S270) gephyrin cluster density was similar to total gephyrin cluster levels after exposure to NMDAR (93%) or GABAaR mAb (99%) (Fig. 2O), indicating that both autoantibodies massively increased the phosphorylation of gephyrin. We further characterised the area of total and phosphorylated (S270) gephyrin clusters, reporting that the decrease in the cluster area of phosphorylated versus total gephyrin was also lost after exposure to NMDAR and GABAaR mAb (Fig. EV2J). Since GABAaR mAb, unlike their NMDAR-targeting counterparts, directly antagonize the ionotropic transmission of GABAaR channels (Kreye et al, 2021), we investigated whether 24 h partial blockade with 5 μM bicuculline—to mimic antibody-induced channel antagonism—was sufficient to elicit receptor and scaffold reorganisation at inhibitory synapses. The GABAaR synaptic content was unaltered (Fig. 2P). Furthermore, bicuculline failed to alter the density of gephyrin scaffolds (Fig. 2Q). Together with the observation that another GABAaR mAb that does not antagonize GABAaR-mediated current produce the same effect as the antagonistic clone (Fig. EV2), it indicates that altered surface dynamics and redistribution of GABAaR is not dependent on its ionotropic modulation by

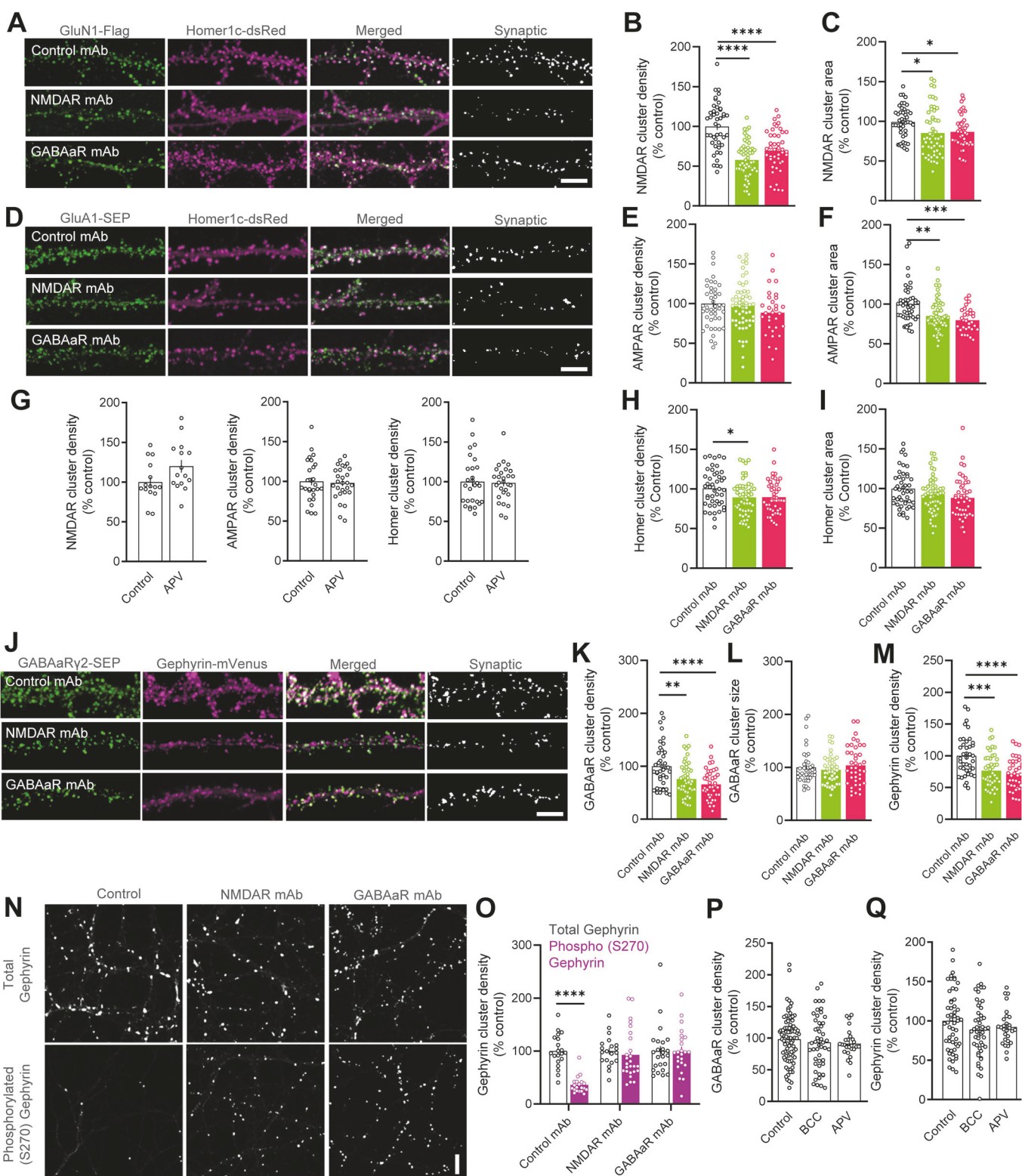

autoantibodies. Altogether, these results demonstrate that NMDAR or GABAaR mAb converge to a similar macroscopic effect likely instigated by the altered surface diffusion and not the lower ionotropic activity, and irrespective to the mAb pharmacological properties.

## NMDAR and GABAaR mAb reduce excitatory and inhibitory synaptic transmission

Based on this in vitro evidence that NMDAR and GABAaR mAb reduce both glutamatergic and GABAergic receptor content at

**Figure 2. NMDAR and GABAaR autoantibodies mutually disrupt excitatory and inhibitory synaptic contents.**

(A) Representative immunostainings of hippocampal neuronal dendrites expressing GluN1-Flag (green) and Homer1c-dsRed (magenta). Binary representations of thresholded colocalisation can be seen as white puncta on the far right. Scale bar = 10 μm. (B, C) Mean synaptic NMDAR cluster density and area, normalised to the Control mAb condition (Control mAb $n = 45$ cells; NMDAR mAb $n = 55$; GABAaR mAb $n = 45$; one-way ANOVA). (D) Representative immunostainings of hippocampal neuronal dendrites expressing GluA1-SEP (green) and Homer1c-dsRed (magenta). Merge of the two signals is shown, in which the colocalisation of AMPAR and Homer1c can be seen as white puncta, and the regions of colocalisation are shown on the far right as binary images. Scale bar = 10 μm. (E, F) Quantified mean per cell of synaptic AMPAR cluster density and area, normalised to the Control mAb condition (Control mAb $n = 46$ cells; NMDAR mAb $n = 57$; GABAaR mAb $n = 31$; one-way ANOVA). (G) From left to right: Quantified means per cell of NMDAR (Control $n = 15$ cells; APV $n = 15$; Student's $t$ test), AMPAR (Control $n = 28$ cells; APV $n = 28$; Student's $t$ test), and homer1c cluster densities (Control $n = 28$ cells; APV $n = 28$; Student's $t$ test), of hippocampal cell dendrites, from cultures exposed to 5 μM of the NMDAR antagonist, APV; or untreated controls. (H, I) Quantified mean per cell of homer1c cluster density and area, normalised to the Control mAb condition (Control mAb $n = 45$ cells; NMDAR mAb $n = 55$; GABAaR mAb $n = 45$; one-way ANOVA). (J) Representative immunostainings of hippocampal neuronal dendrites expressing γ2-SEP (green) and gephyrin-mVenus (magenta). Binary representations of thresholded colocalisation can be seen as white puncta on the far right. Scale bar = 10 μm. (K, L) Mean synaptic GABAaR cluster density and area, normalised to the Control mAb condition (Control mAb $n = 40$ cells; NMDAR mAb $n = 43$; GABAaR mAb $n = 41$; one-way ANOVA). (M) Mean gephyrin cluster density, normalised to the Control mAb condition (Control mAb $n = 40$ cells; NMDAR mAb $n = 43$ cells; GABAaR mAb $n = 41$ cells; one-way ANOVA). (N) Representative immunostainings of total gephyrin puncta and gephyrin phosphorylated at S270. Scale bar = 20 μm. (O) Mean total versus phosphorylated gephyrin puncta density, normalised to the level of total gephyrin staining (Control mAb total gephyrin $n = 20$ cells; Control mAb phospho-gephyrin $n = 20$; NMDAR mAb total gephyrin $n = 20$; NMDAR mAb phospho-gephyrin $n = 26$; GABAaR mAb total gephyrin $n = 24$; GABAaR mAb phospho-gephyrin $n = 24$; multiple $t$ tests with Benjamini and Hochberg correction for false discovery rate). (P, Q) Quantified means per cell of synaptic GABAaR (Control $n = 81$ cells; BCC $n = 49$; APV $n = 28$; one-way ANOVA) and gephyrin cluster densities (Control $n = 81$ cells; BCC $n = 49$; APV $n = 28$; one-way ANOVA), of hippocampal cell dendrites, from cultures exposed to 1 μM of the GABAaR antagonist, bicuculline (BCC); 5 μM of the NMDAR antagonist, APV; or untreated controls. Data information: All error bars represent the standard error of the mean. Significance levels are represented as *$P < 0.05$, **$P < 0.01$, ***$P < 0.001$ and ****$P < 0.0001$.

synapses, we next investigated the fast transmissions in preserved hippocampal networks. Both excitatory and inhibitory synaptic currents were revealed using whole-cell patch-clamp electrophysiology in a voltage-clamp configuration. Hippocampal slices were exposed to NMDAR or GABAaR mAb autoantibody for 24 h prior to recording. The voltage clamping at −70 mV was performed for 10 min to uncover AMPAR-mediated spontaneous excitatory postsynaptic currents (sEPSC, Fig. 3A,B). After which, cells were clamped at 0 mV to reveal GABAaR-mediated spontaneous inhibitory postsynaptic currents (sIPSC, Fig. 3E,F). The sEPSC amplitude was significantly reduced after exposure to NMDAR or GABAaR mAb (Fig. 3C,D). There was no change in the frequency and either rise or decay kinetics of sEPSC of either pathogenic antibody (Fig. EV3A–C). This suggests that the loss of AMPARs from excitatory synaptic compartments is not selective for particular subunit containing channels, instead reflecting an overall decrease in synaptic AMPARs. Quantification of sIPSC amplitudes revealed a similar impact on inhibitory signalling by NMDAR or GABAaR mAb. Compared to control, we observed a significant reduction after exposure to both NMDAR and GABAaR mAb (Fig. 3G,H), with a 16% reduction in frequency after exposure to either mAb. However, this trend failed to reach significance and we found no impact on event kinetics (Fig. EV3D–F). The reduction in GABAergic signalling is expected by GABAaR mAb exposure since the mechanisms of action of these antibodies is, in part, a direct antagonism of GABAaR channel function. However, the impact of NMDAR mAb on GABAergic fast transmission further indicate the effective crosstalk induced by autoantibodies. To compare the E/I balance of the synaptic inputs, we expressed the mean amplitudes of the AMPAR-mediated sEPSCs relative to the mean amplitudes of the GABAaR-mediated sIPSC for each recorded cell. This revealed a significant shift towards hyperexcitation in the presence of NMDAR (155%) and GABAaR mAb (157%) relative to the control mAb condition (Fig. 3I). This demonstrates that although there is a general reduction of synaptic inputs to CA1 pyramidal cells, the deficits in GABAergic transmission are consistently greater in magnitude compared to the impacts on glutamatergic transmission. The ionotropic transmissions of both NMDA- and GABAaR are

not confined solely to synaptic compartments, but also to tonic excitatory and inhibitory currents through extrasynaptically recruited receptors. We thus measured tonic GABAergic current on CA1 pyramidal neurons after exposure to autoantibody samples. Whole-cell patch-clamp recordings were collected in the presence of tetrodotoxin. After a stable baseline, current was recorded for 5 min, bicuculline was added into the extracellular solution to reveal the tonic GABAaR-mediated inhibitory current, apparent as a downward deflection in the trace baseline (Fig. 3J). Analysis of baseline shift in hippocampal cells exposed to control antibody, revealed a tonic GABAergic current of ~18 pA in amplitude (Fig. 3K,L). Exposure to NMDAR mAb significantly reduced this tonic inhibitory current amplitude, whereas exposure to GABAaR mAb virtually abolished the GABAergic tonic current (Fig. 3K,L). Altogether, these recordings demonstrate that NMDAR or GABAaR mAb induce comparable reduction in fast excitatory and inhibitory transmissions and shift the E/I balance toward excitation.

## Excitatory and inhibitory synaptic crosstalk is activity-dependent

Given the synaptic content of glutamatergic and GABAa receptors is modulated by hippocampal network activity (Hennequin et al, 2017; Turrigiano, 2017), we investigated whether neuronal activity plays a role in the crosstalk effect of pathogenic autoantibodies. Hippocampal neurons were exposed 24 h with control, NMDAR or GABAaR mAb, either in standard cell culture medium or medium supplemented with 20 nM TTX to block network activity (Fig. EV4A). NMDAR mAb decreased NMDAR cluster density in the absence or presence of TTX (Fig. EV4B), indicating that the membrane disorganisation of NMDARs by NMDAR mAb is activity-independent. However, the GABAaR cluster density was not longer altered by NMDAR mAb in the presence of TTX (Fig. EV4C). These data indicate that the crosstalk effects of NMDAR mAb on inhibitory synapses rely on neuronal activity. Remarkably, we observed the mirror effect of GABAaR mAb on excitatory synapses. Indeed, GABAaR mAb reduce GABAaR cluster density

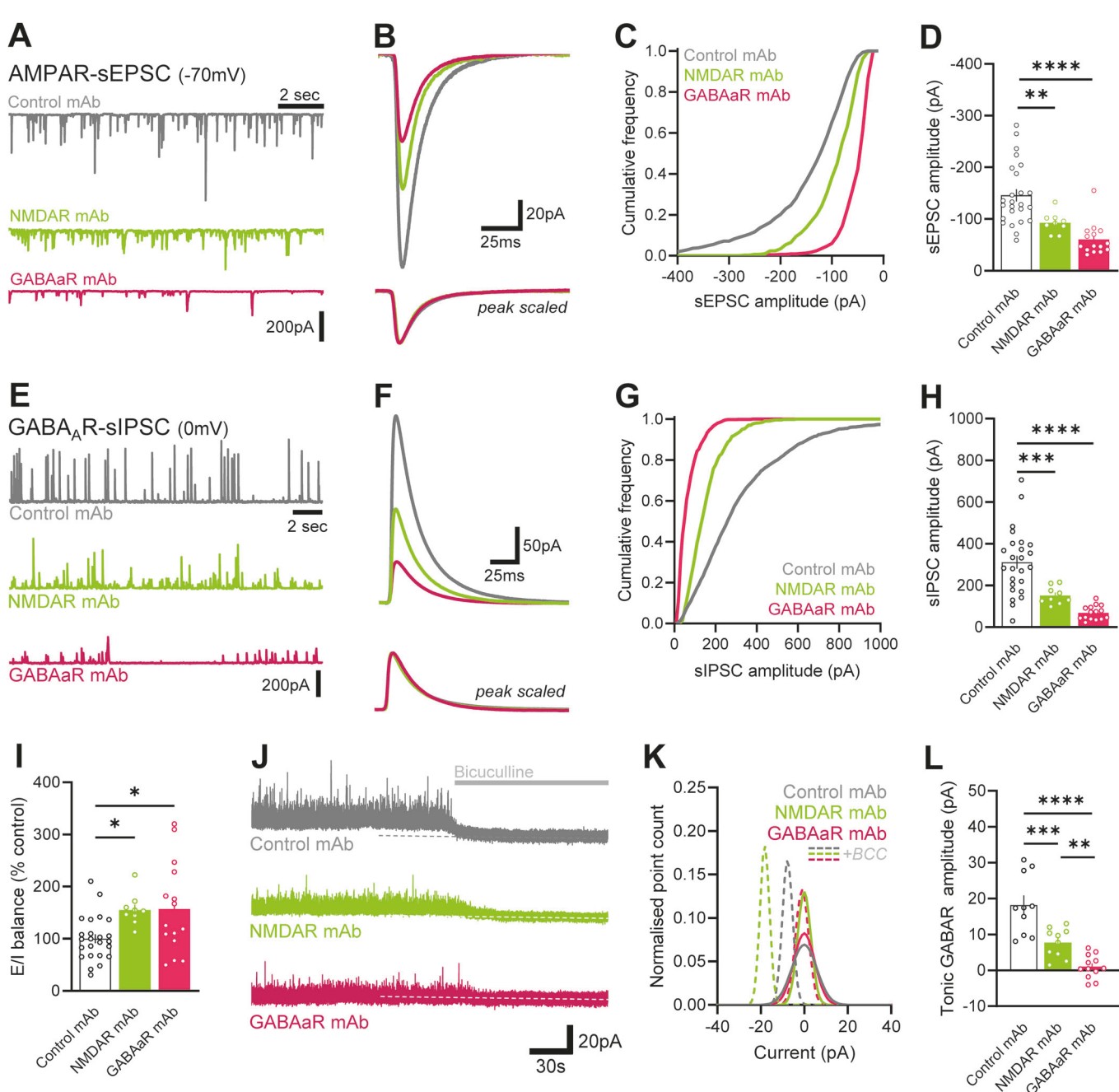

**Figure 3.  Autoantibodies mutually dysregulate spontaneous excitatory and inhibitory neurotransmission.**

(A, B) Representative trace recordings from CA1 principal cells in organotypic hippocampal slices and mean event traces of AMPAR-mediated spontaneous excitatory postsynaptic currents, AMPAR-sEPSC. (C, D) Cumulative frequency distributions of AMPAR-sEPSC amplitudes (Control mAb $n = 1020$ events; NMDAR mAb $n = 296$; GABAaR mAb $n = 449$) and mean amplitude (Control mAb $n = 25$ cells; NMDAR mAb $n = 9$; GABAaR mAb $n = 15$; one-way ANOVA). (E, F) Representative trace recordings from CA1 principal cells in organotypic hippocampal slices and mean event traces of GABAaR-mediated spontaneous inhibitory postsynaptic currents, GABAaR-sIPSC. (G, H) Cumulative frequency distributions (Control mAb $n = 1952$ events; NMDAR mAb $n = 538$; GABAaR mAb $n = 873$) of GABAaR-sIPSC amplitudes and mean amplitude (Control mAb $n = 25$ cells; NMDAR mAb $n = 9$; GABAaR mAb $n = 15$; one-way ANOVA). (I) Excitatory-inhibitory balance quantifications resulting from mean cellular AMPAR-sEPSC amplitude divided by mean cellular GABAaR-sIPSC amplitudes (Control mAb $n = 24$ cells; NMDAR mAb $n = 9$; GABAaR mAb $n = 15$; one-way ANOVA). (J) Representative trace recordings of GABAaR-sIPSC before and after wash-in of 20 µM of GABAaR antagonist, bicuculline (BCC). (K) Frequency distributions of the normalised point count, generated from event-free baselines before and after BCC application, and (L) mean cellular amplitude of the recorded tonic GABAergic current (Control mAb $n = 10$ cells; NMDAR mAb $n = 10$; GABAaR mAb $n = 12$; one-way ANOVA). Data information: All error bars represent the standard error of the mean. Significance levels are represented as *$P < 0.05$, **$P < 0.01$, ***$P < 0.001$ and ****$P < 0.0001$.

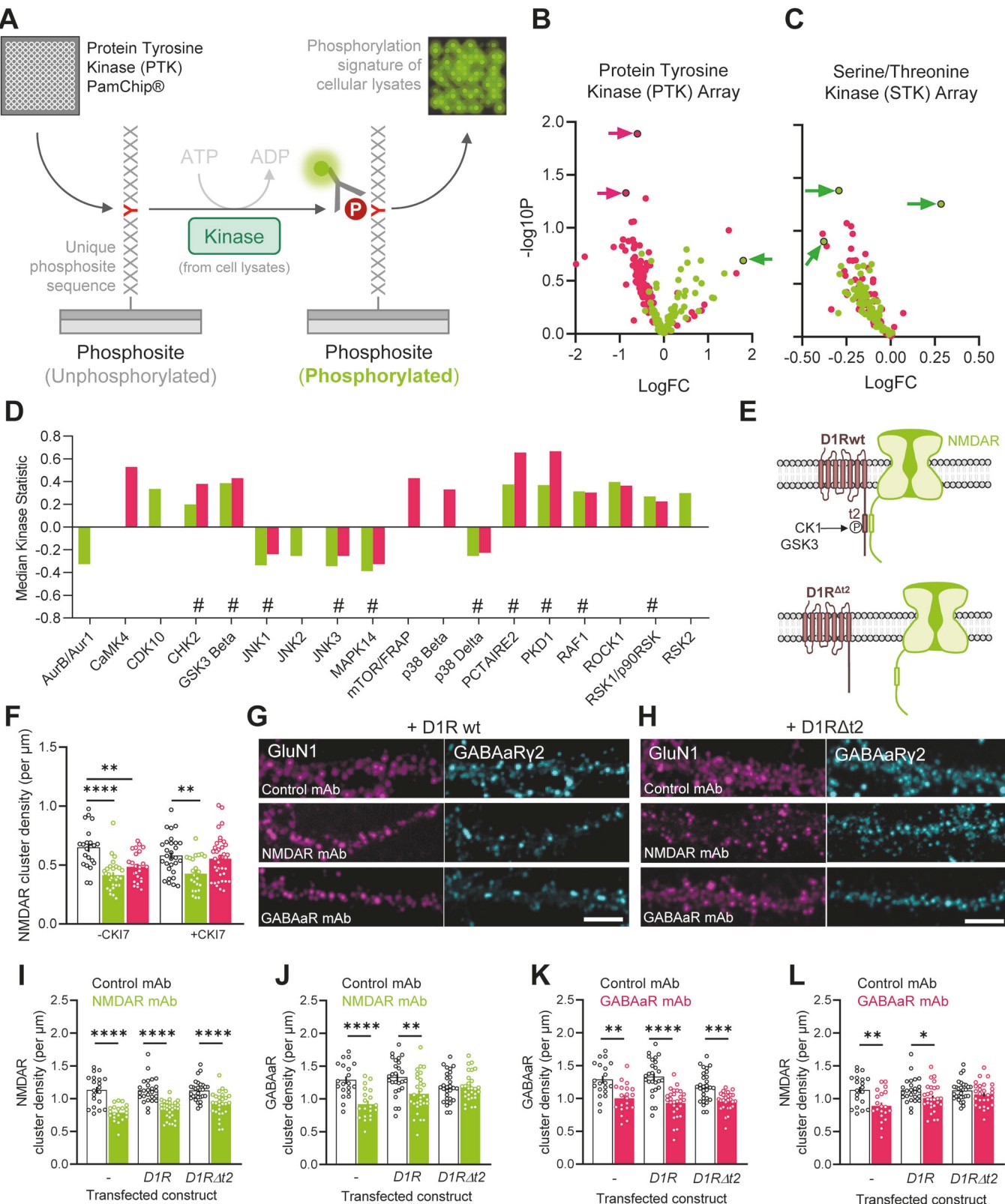

◀  **Figure 4.  Intracellular signalling cascades and surface protein interactions are dysregulated by NMDAR and GABAaR autoantibody exposure.**

(A) Schematic representation of experimental protocols employed in the microarray kinase assay. (B, C) Volcano plots of the quantified fluorescence change of individual phosphosites after processing of cell lysates from hippocampal cultures exposed to NMDAR mAb (green) or GABAaR mAb (pink) relative to Control mAb treated cultures using STK or PTK array. Arrowheads indicate phosphosites that show significant difference compared to control ($P < 0.05$). Results are expressed by plotting the effect area (x-axis, LFC or delta) versus significance ($y$ axis, -log10($P$ value)) of the test. (D) Upstream Kinase Analysis (UKA) to predict differential kinase activity in the test condition compared to the control based on the phosphorylation of a set of phosphosites, "#" represents cases in which significant alterations were identified after exposure to both NMDAR mAb (green) and GABAaR mAb (pink) compared to a control mAb. (E) Schematic illustration showing the loss of interaction between NMDAR and D1R after introduction of a deletion on the D1R C-terminal domain, D1RΔt2. (F) Cell mean quantification of synaptic NMDAR cluster density after 24 h exposure to Control mAb, NMDAR mAb or GABAaR mAb, in the presence or absence of CK1 antagonist, CKI (No CKI: Control mAb $n = 22$ cells; NMDAR mAb $n = 28$; GABAaR mAb $n = 24$; Plus CKI: Control mAb $n = 29$; NMDAR mAb $n = 23$; GABAaR mAb $n = 37$; one-way ANOVA). (G, H) Representative immunostainings of hippocampal neurons expressing GluN1 (magenta) and γ2 (cyan), with wild-type D1R or (H) D1R Δt2. Scale bar = 10 μm. (I, J) Mean dendritic NMDAR and (J) GABAaR cluster density after exposure to Control mAb and NMDAR mAb, with coexpression of endogenous-only (−), wild-type D1R or D1R Δt2 (untransfected: Control mAb $n = 20$ cells; NMDAR mAb $n = 20$; D1R-WT: Control mAb $n = 26$; NMDAR mAb $n = 29$; D1R$^{Δt2}$: Control mAb $n = 29$; NMDAR mAb $n = 28$). (K, L) Quantified cell means of dendritic GABAaR and NMDAR cluster density after exposure to Control mAb and GABAaR mAb, with coexpression of endogenous-only (−), wild-type D1R or D1R Δt2 (untransfected: Control mAb $n = 20$ cells; GABAaR mAb $n = 20$; D1R-WT: Control mAb $n = 26$; GABAaR mAb $n = 28$; D1R$^{Δt2}$: Control mAb $n = 29$; GABAaR mAb $n = 27$). Data information: All error bars represent the standard error of the mean. Significance levels are represented as *$P < 0.05$, **$P < 0.01$, ***$P < 0.001$ and ****$P < 0.0001$.

regardless of TTX exposure (Fig. EV4D). Yet, the NMDAR cluster density was not anymore altered by GABAaR mAb in the presence of TTX (Fig. EV4B). Thus, the crosstalk effect of GABAaR mAb on NMDAR is activity-dependent. These findings demonstrate a biphasic effect of autoantibodies.

## Multiple protein kinase activity changes following exposure to autoantibodies

In order to identify the putative signalling pathway(s) underpinning the crosstalk effects of autoantibodies, we focused our attention to protein kinase activity in the presence of NMDAR and GABAaR mAb. Indeed, synaptic activity is known to alter various protein kinase and phosphatase activities and our observation that gephyrin phosphorylation drastically changed following exposure to autoantibodies prompt us to measure protein kinase activity. To identify candidate protein kinases, we used a kinase screening assay based on fluorescent labelling of phosphorylated substrates developed with microarray technology (PamGene, Fig. 4A). Neuronal lysates, collected from cultured neurons exposed to control, NMDAR or GABAaR mAb were screened for changes in activity of both tyrosine (PTK) and serine/threonine-specific (STK) protein kinases. We identified 1 (PTK) and 3 (STK) phosphosites that were significantly altered after exposure to NMDAR mAb, and 2 (PTK) phosphosites that were significantly altered after exposure to GABAaR mAb (Fig. 4B,C). These substrates identify then less than twenty protein kinases whose activities have been significantly altered by NMDAR or GABAaR mAb. Protein kinase whose activities were downregulated by both autoantibodies are: TEK Receptor Tyrosine Kinase (TEK), Focal adhesion kinase 2 (FAK2), Spleen tyrosine kinase (SYK), c-Jun N-terminal kinases 1 & 2 (JNK1&2) and p38δ mitogen-activated protein kinase (p38δ) (Fig. 4D). Conversely, upregulated protein kinases included: Fibroblast growth factor receptor 4 (FGFR4), Receptor tyrosine-protein kinase (RET), Protein kinase D1 (PKD1), Checkpoint kinase 2 (CHK2) and Glycogen synthase kinase-3 beta (GSK3β) (Fig. 4D). Intriguingly, of these identified kinases, GSK3β is known to phosphorylate gephyrin at S270. As such, it is tempting to speculate that the upregulation of this kinase, induced by both NMDA and GABAaR mAb, may pose a likely candidate mechanism for the selective loss of inhibitory synapses. Furthermore, it is evident that both NMDAR and GABAaR mAb elicit

substantial dysregulations of intracellular signalling cascades, with 11 out of 18 protein kinases (~60%) whose activities were similarly altered by both autoantibodies. Together, these data show a certain degree of convergence in intracellular signalling cascades between the autoantibodies, suggesting that intracellular kinase-mediated pathways likely form a complex substrate of altered signalling.

## Dopamine-NMDA receptor interaction contributes to the autoantibody-mediated crosstalk effect

Among the various intracellular cascades that can tune the membrane redistribution of NMDAR and GABAaR following exposure to autoantibodies, the direct protein–protein interaction between transmembrane receptors is of prime interest (Borroto-Escuela et al, 2017; Petit-Pedrol and Groc, 2021). In this context, dopamine receptors are of particular interest since (i) they directly interact with both NMDAR and GABAaR, (ii) their interaction is regulated phosphorylation and (iii) GSK3, which activity is upregulated by both autoantibodies, forms a complex with Casein kinase 1 (CK1), a known modulator of the NMDAR/D1R interaction (Fig. 4E) (Li et al, 2020; Lee et al, 2002; Petit-Pedrol and Groc, 2021). Although the expression of D1R is heterogenous across (ventral vs dorsal) and within (areas) hippocampi (Edelmann and Lessmann, 2018), we here used cultured hippocampal neurons from the whole hippocampus to ensure a rather large proportion of D1R-positive neurons. We hypothesised that if D1R-GluN1 interaction plays a role in linking the neuronal-activity-modulated intracellular signalling cascades with surface organisation, then abolition of this interaction should in turn prevent the cross-organisational impacts of NMDAR and GABAaR mAb on GABAaR and NMDAR, respectively. First, an inhibitor of the CK1, the CKI7 was applied together with the autoantibodies and observed that the effect of the GABAaR mAb on NMDAR was lost when CKI7 was present (Fig. 4F). To manipulate the D1R-GluN1 interaction, we employed a genetic strategy, in which neuronal cultures were transfected with either a WT D1R-expressing construct, or a D1R$^{Δt2}$-expressing construct, in which the GluN1 interacting peptide sequence is deleted from the intracellular tail. As a result, NMDAR and D1R$^{Δt2}$ are unable to interact (Fig. EV4E). In these experiments, exposure to mAb for 24 h showed that NMDAR mAb decrease dendritic NMDAR cluster density, irrespective of the interaction with D1R (Fig. 4F–I).

However, when the D1R-GluN1 interaction is lost, in cultures transfected with D1R$^{\Delta t2}$, NMDAR mAb did not cause an effect on the cluster density of GABAaR (Fig. 4J). Thus, our observations suggest that D1R-GluN1 interaction may play a role in mediating the cross-effect of NMDAR mAb on GABAaR disorganisation. Similarly, while the effect of GABAaR mAb on GABAaR cluster density is preserved in neurons expressing WT D1R, the cross-organisational impact on NMDAR cluster density along the dendritic compartments is abolished (Fig. 4K,L). Quantification of the expression of the exogenously expressed D1R or D1R$^{\Delta t2}$ demonstrated that the observed changes were not due to a different cell-surface expression but a displacement of the receptors (Fig. EV4F–H). While we do not rule out the possibility that other surface proteins—and their interactions with NMDAR or GABAaR—will also likely be modulated by the identified intracellular signalling dysregulations, it appears that D1R interaction with NMDAR may play a substantial role in the orchestration of membrane-wide disorganisation of NMDAR and GABAaR after exposure to GABAaR and NMDAR mAb, respectively. It will also be of interest to test the implication of other members of the dopamine receptor family as, for instance, dopamine D5 receptor expression is higher than that of D1R in the hippocampus.

## GABAaR mAb increase principal cell excitability

Given membrane receptor disorganisation and synaptic transmission deficits, we tested whether autoantibodies also have the potency to alter intrinsic properties of hippocampal principal cells. We used a whole-cell current-clamp protocol, in which CA1 neurons were administered a series of hyper-to-depolarising current injections (from −150 to 400 pA in 50 pA increments) (Fig. 5A–C). Input–output curves were generated, indicating an increased propensity for action potential firing after exposure to GABAaR mAb, but not NMDAR mAb (Fig. 5D). Further, exposure to NMDAR mAb had no impact on the resting membrane potential (−61 mV), whereas GABAaR mAb elicited a depolarisation of the resting state potential (−48 mV; Fig. 5E). Given this shift in the resting potential, we also characterised the threshold for action potential discharge and rheobase. None of autoantibodies altered the threshold for action potential (Fig. EV3G), whereas the rheobase was shifted in favour of hyperexcitation after exposure to GABAaR mAb (41 pA) (Fig. 5F), which is expected given the depolarised resting potential and unchanged threshold for action potential generation. In addition, we observed a significant reduction in action potential amplitudes and an increase in their half-width (Fig. EV3J–L), which may be explained by an alteration in GABAaR-mediated signalling at the axon initial segment. Taken together, these findings suggest that GABAaR mAb, but not NMDAR mAb, alter the intrinsic excitability of principal cells in the hippocampus. The input resistance was also impacted by GABAaR mAb (Fig. 5G). Altogether, these data indicate that GABAaR mAb alter the intrinsic properties of CA1 principal cells, favouring their excitability, whereas NMDAR mAb was without effect.

## Both NMDAR and GABAaR mAb increase principal cell network activity

Because autoantibodies alter excitatory and inhibitory synaptic transmissions and, in part, cell-intrinsic properties, an alteration of the principal cell network activity is possible. More specifically, one may predict that i) NMDAR mAb will increase principal cell firing and network activity since they increase E/I balance of inputs and decrease tonic GABAergic current, and ii) GABAaR mAb will also increase principal cell firing and network activity since they increase E/I balance of inputs, decrease tonic GABAergic current, and increase principal cell excitability. To directly test the effect of NMDAR and GABAaR mAb on principal cell firing and network activity, we recorded cells in a cell-attached voltage-clamp mode, quantifying the spiking with a minimally invasive recording configuration. Both NMDAR and GABAaR mAb increased the firing of principal cells (Fig. 5H,I). To further address this question and explore the network activity, we employed a calcium imaging approach as a proxy for action potential discharges. CA1 principal cells of hippocampal slices were virally transfected with the calcium sensor, GCaMP6, to visualise somatic calcium influx and cell firing (Fig. 5J). Consistently, the frequency of spontaneous calcium transients was significantly increased (Fig. 5K), indicating that exposure to either NMDAR or GABAaR mAb alone is sufficient to drive network hyperactivation. To further understand if this hyperactive state is driven by a generalised increase in principal cell activity, or by a select subgroup of hyperactive neurons that in turn evoke network effects, we analysed the relative frequency distributions and maximum firing frequency of individual cells within each recorded slice. We found that the cumulative distributions of cellular activity are right-shifted by NMDAR and GABAaR mAb, indicating an overall increase in cellular activity (Fig. EV5A). In addition, we did not observe an alteration of the maximum firing frequency after exposure to either autoantibody (Fig. EV5B), suggesting that this increase is relatively homogeneous. Recent reports have indicated that NMDAR mAb induce a hypersynchronous network state between excitatory neurons in hippocampal circuits (Ceanga et al, 2023). To confirm this observation and explore whether it also applies for GABAaR mAb, we calculated the mean correlation index (Wong et al, 1993) from pairwise comparisons between all cells in the recorded networks. We uncovered a significant increase in the mean cell correlation index after exposure to NMDAR mAb, but only a trend for GABAaR mAb (Fig. EV5C). Analysis of simultaneously occurring cellular firing, here referring to calcium transients identified within the same imaging frame (300 ms), revealed that both NMDAR and GABAaR mAb increase this rate (Fig. EV5D). Finally, we investigated the percentage of pairwise comparisons that yielded a positive correlation index, across all pairs of neurons. This analysis revealed a significant increase in the percentage of positively correlated spike trains after exposure to NMDAR mAb, and only a trend for the GABAaR mAb group (Fig. 5L). Altogether, these data demonstrated that both NMDAR and GABAaR mAb increase the firing rate of principal cells as well as CA1 principal cell network activity, which is further synchronised by NMDAR mAb.

## NMDAR mAb decrease interneuron excitability and silence their firing

The activity of the hippocampal neuronal network is highly tuned by interneurons (Topolnik and Tamboli, 2022). To grasp a comprehensive view of the impact of autoantibodies on the hippocampal network, we thus recorded intrinsic properties and

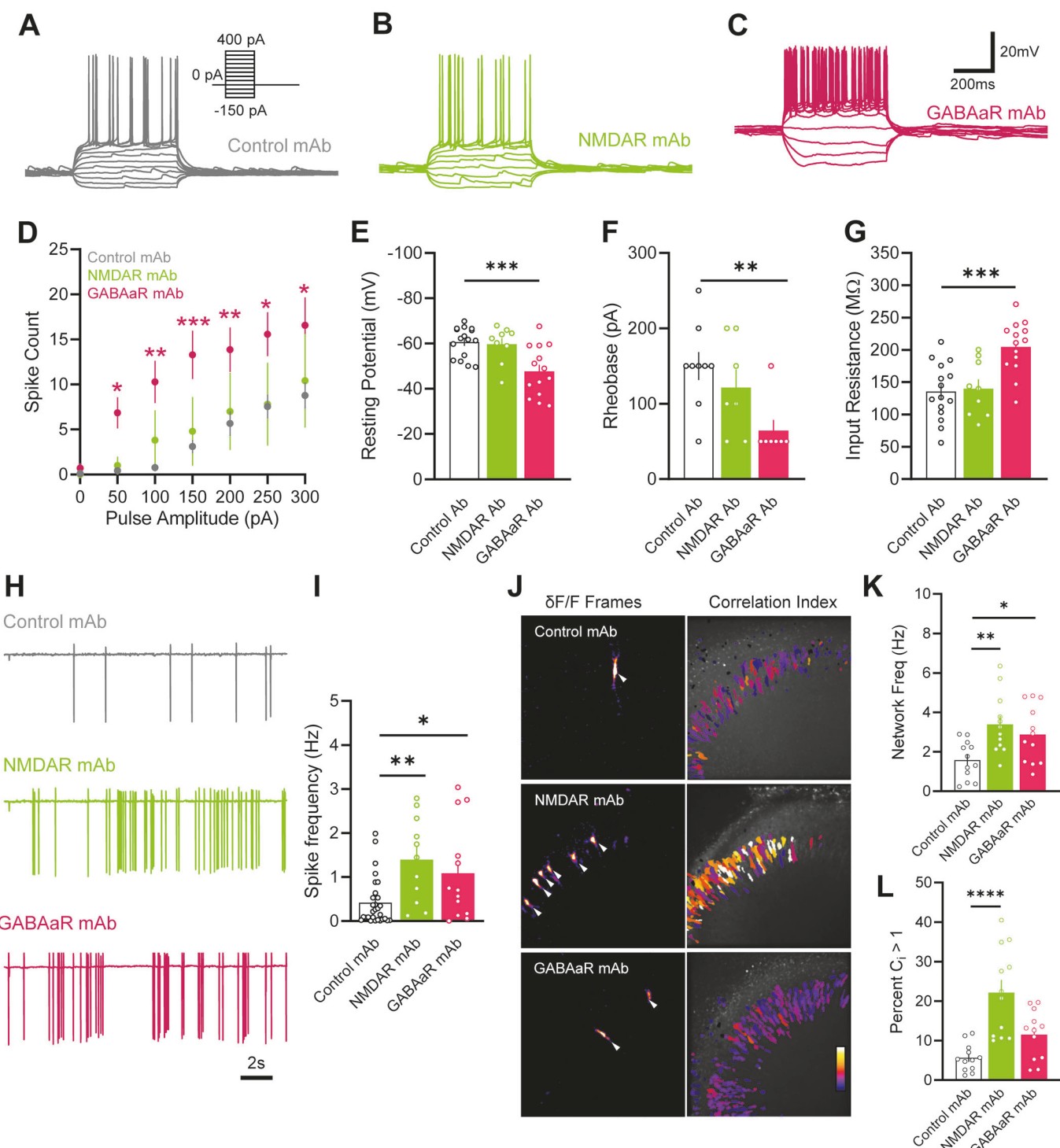

**Figure 5. Autoantibodies alter cell-intrinsic properties and principal cell network activity.**

(A–C) Representative current-clamp recordings from CA1 principal cells in organotypic hippocampal slices exposed to Control mAb, NMDAR mAb and GABAaR mAb. (D) Input–output quantification of mean number of action potential discharges elicited in response to depolarising current injections (Control mAb $n = 9$ cells; NMDAR mAb $n = 7$; GABAaR mAb $n = 7$; one-way ANOVA). (E, F) Mean resting membrane potential and (F) rheobase of CA1 pyramidal neurons (Control mAb $n = 9$ cells; NMDAR mAb $n = 7$ cells; GABAaR mAb $n = 7$ cells; one-way ANOVA). (G) Mean input resistance of CA1 pyramidal neurons (Control mAb $n = 15$ cells; NMDAR mAb $n = 9$; GABAaR mAb $n = 14$; one-way ANOVA). (H, I) Representative trace recordings of CA1 pyramidal neurons in cell-attached voltage-clamp configuration, after exposure to Control, NMDAR or GABAaR mAb and the mean spontaneous spike frequency (Control mAb $n = 29$ cells; NMDAR mAb $n = 11$; GABAaR mAb $n = 13$; one-way ANOVA). (J) Representative frames taken from post-processed calcium imaging acquisitions, and the mean cellular correlation indices. (K, L) Mean network frequency from recorded CA1 hippocampal networks daniel the mean percentage of thresholded correlation indices (Control mAb $n = 12$ slices; NMDAR mAb $n = 12$; GABAaR mAb $n = 12$; one-way ANOVA). Data information: All error bars represent the standard error of the mean. Significance levels are represented as $*P < 0.05$, $**P < 0.01$, $***P < 0.001$ and $****P < 0.0001$.

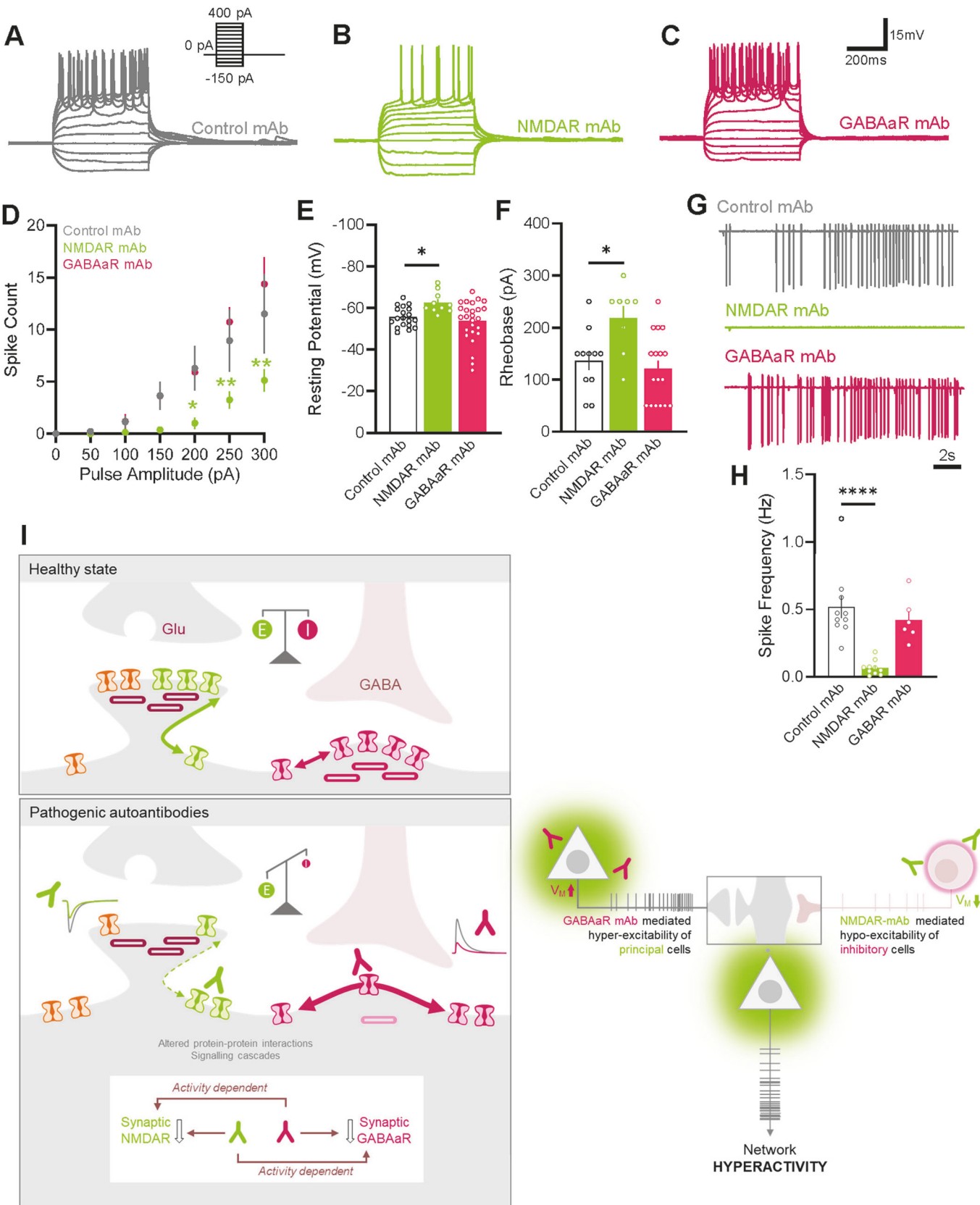

**Figure 6. NMDAR mAb decrease interneuron excitability and silence their firing.**

(A–C) Representative current-clamp recordings from CA1 inhibitory interneurons in organotypic hippocampal slices exposed to Control mAb (A), NMDAR mAb (B) and GABAaR mAb (C). (D) Input–output quantification of mean number of action potential discharges elicited in response to depolarising current injections (Control mAb $n = 14$ cells; NMDAR mAb $n = 8$; GABAaR mAb $n = 11$; one-way ANOVA). (E, F) Mean resting membrane potential (Control mAb $N = 11$ cells; NMDAR mAb $N = 8$; GABAaR mAb $N = 17$; one-way ANOVA) and rheobase of CA1 pyramidal neurons (Control mAb $n = 19$ cells; NMDAR mAb $n = 10$; GABAaR mAb $n = 26$; one-way ANOVA). (G, H) Representative trace recordings of CA1 inhibitory interneurons in cell-attached voltage-clamp configuration, after exposure to Control, NMDAR or GABAaR mAb and the mean spontaneous spike frequency (Control mAb $n = 10$ cells; NMDAR mAb $n = 11$; GABAaR mAb $n = 6$; one-way ANOVA). (I) Left: Schematic representation of synaptic alterations in the context of NMDAR and GABAaR mAb exposure, specifically demonstrating the depletion of AMPAR, NMDARs and GABAaRs from synaptic compartments, altered mobility of remaining NMDARs and GABAaRs, resulting in disrupted EI balance of synaptic inputs. Right: Schematic representation of antibody selective alterations of principal (triangular) and inhibitory neuron (circular) intrinsic excitability, and resulting overall network hyperactivity. Data information: All error bars represent the standard error of the mean. Significance levels are represented as *$P < 0.05$, **$P < 0.01$, and ****$P < 0.0001$.

firing rate of CA1 stratum radiatum interneurons exposed to either NMDAR or GABAaR mAb. Recording hyper- to depolarising current steps to interneurons revealed a significant decrease in the propensity for action potential generation in response to depolarising current injection in hippocampal slices exposed to NMDAR mAb, but not GABAaR mAb (Fig. 6A–D). Moreover, we identified a modest but significant hyperpolarisation of the resting membrane potential in interneurons exposed to NMDAR mAb (Fig. 6E). Since we did not observe any change in the threshold for action potential generation in these cells after exposure to either NMDAR or GABAaR mAb (Fig. EV3H,I), we anticipate that this deflection in resting potential will serve to increase the requirement for excitatory input, before successful elicitation of neuronal firing in interneurons. This expectation was supported by the increased rheobase in interneurons exposed to NMDAR mAb (Fig. 6F). Consistent with the above data, we identified a massive loss of interneuron firing after exposure to NMDAR mAb using a cell-attached voltage-clamp approach (Fig. 6G,H). This reduction in inhibitory interneuron activity was not observed after exposure to GABAaR mAb, suggesting that this silencing of GABAergic activity is a unique pathology related to NMDAR mAb. Collectively, these data demonstrated that NMDAR mAb, but not GABAaR mAb, downregulate the interneuron excitability and massively decrease their firing within the hippocampal CA1 network. Taken in the context of our existing findings, we would postulate that this increase in excitatory output is primarily driven by a shift towards hyperexcitation of phasic synaptic inputs, that is compounded either by an increase in intrinsic excitability by GABAaR mAb; or by the ablated inhibitory networks in the case of NMDAR mAb exposure, which then fail to control excitatory connections between principal cells, resulting the hypersynchronous network phenotype we observed after exposure to NMDAR mAb (Fig. 6I).

## Discussion

How autoantibodies targeting neurotransmitter receptors mediating either excitation or inhibition lead to similar clinical symptoms and behavioural deficits recently emerged as a challenging question. In this study, we demonstrate that autoantibodies from patients with anti-NMDAR or anti-GABAaR encephalitis impact the membrane diffusion and distribution of glutamatergic and GABAergic receptors, resulting in deficits of both excitatory and inhibitory synaptic drive. Indeed, both NMDAR and GABAaR mAb drive alterations in the surface diffusion of their target antigens, as expected from previous studies using patient-derived

biological samples (Jezequel et al, 2017; Mikasova et al, 2012). Exposure to NMDAR mAb reduces the surface diffusion of NMDAR at excitatory synapses after 24 h, suggesting reduced membrane dynamics of remaining synaptic NMDARs after long exposure to NMDAR mAb. On the other hand, GABAaR mAb induce an increase in surface diffusion of GABAaR at inhibitory synapses, likely dependent on the loss of the postsynaptic scaffold gephyrin in these synapses. Surprisingly, we uncovered that both NMDAR- and GABAaR mAb also provoke similar alterations in the surface diffusion of GABAaR and NMDAR, respectively. Autoantibody exposure, irrespective of the target antigen, impedes the correct localisation of receptors to synaptic compartments and induce a deficit in synaptic transmissions. These effects were independent on the properties of the mAb since, for instance, GABAaR mAb that do inhibit GABAaR-mediated current or GABAaR mAb that do not inhibit this current produced the same cross-effect. Consistently, blocking activity of the NMDAR or GABAaR using antagonists did not phenocopy the effects on receptor clusters, suggesting that the mechanical disorganisation of the receptor by the mAb is responsible for the effects. Analysis of spontaneous AMPAR-mediated excitatory postsynaptic currents revealed that both NMDAR and GABAaR mAb reduce excitatory input onto CA1 hippocampal pyramidal neurons. Remarkably, despite the relatively moderate macroscale disorganisation of AMPAR at excitatory synapses, the larger functional reduction in excitatory current amplitudes would suggest that autoantibodies elicit a substantial disruption of nanoscale synaptic architecture (Jezequel et al, 2017; Ladepeche et al, 2018), possibly through a destabilisation of nanocolumns, in which AMPAR clusters are closely aligned with presynaptic glutamate release sites (Nair et al, 2013; Tang et al, 2016). Moreover, GABAaR-mediated inhibitory signalling was substantially reduced by both NMDAR and GABAaR mAb. The magnitude of reduction in inhibitory transmission was greater than that observed in excitatory transmission, yielding a clear shift in the excitation-inhibition balance of synaptic inputs in favour of hyperexcitation. These homogenous impacts, by both NMDAR and GABAaR mAb, suggest convergent routes towards pathogenesis across distinct autoimmune encephalitides. Consistent with the fact that neuronal activity modulates the surface expression, trafficking and organisation of many synaptic receptors, we observed that blockade of neuronal activity with TTX application during mAb exposure maintained the direct antigen effects of both pathogenic mAb but this manipulation abrogated their cross-system effects. The mAb alter the activity of an array of protein kinases, with a high degree of convergence between autoantibodies. Given that a high degree of kinase regulatory processes are intricately linked with neuronal

activity, it is tempting to speculate that convergent network hyperactivity—induced by both NMDAR and GABAaR mAb—lies upstream of kinase pathway dysfunction. As such, blockade of network and neuronal activity by TTX may abrogate the cross-synaptic effects by impeding pathological alterations in kinase activity. Among the putative mechanisms that could be affected, we show that the protein–protein interaction between NMDAR and dopamine receptors contribute to the cross-effects likely through the regulation of the interaction by protein kinases. Whether the altered dopamine signalling induced by encephalitis autoantibodies (Carceles-Cordon et al, 2020; Grea et al, 2017) contribute the psychiatric and/or neurological (e.g., dyskinesia) features remain to be addressed.

The major alterations of synaptic inputs are compounded by shifts in intrinsic cellular excitability of pyramidal neurons exposed to GABAaR mAb. While this shift towards hyperexcitability was not observed in principal cells exposed to NMDAR mAb, an opposing reduction in excitability of interneurons was observed. One may propose that this bidirectional action on opposing neuronal systems yields the same endpoint of increased hippocampal activity at the network scale. A common overarching hyperactivation of hippocampal pyramidal cells, culminating from the dysregulation across the synaptic and cellular functions. Interestingly, network hypersynchrony induced by NMDAR mAb has recently also been described in hippocampal *ex vivo* recordings in a passive-transfer mouse model of NMDAR encephalitis and in further recent studies (Andrzejak et al, 2022; Ceanga et al, 2023). Here, we found severely reduced excitability in hippocampal interneurons by NMDAR mAb which might serve as a potential driver of network hyperexcitability (Andrzejak et al, 2022; Kreye et al, 2021; van Casteren et al, 2022; Wright et al, 2021). Although the molecular mechanisms underpinning these cell-intrinsic property changes remain to be defined, one may envision that autoantibodies alter the membrane organisation and trafficking of ion channels controlling the cell-intrinsic properties. Antibody-based cross-linking of NMDAR, which could resemble to some aspect the impact of NMDAR-Abs, alter the firing pattern of dopamine neurons from the ventral tegmental area (Etchepare et al, 2021). Although a dysfunctional crosstalk between NMDAR and SK potassium channels is suspected in this specific phenotype, the functional interplay between NMDAR and various ion channels and transporters, e.g., Kv4.2 (Jung et al, 2008), fuel the hypothesis that autoantibodies against neurotransmitter receptors can indirectly alter ion channel membrane content and distribution, and consequently impair cell-intrinsic properties. This also suggests that autoimmune disruption of neuronal surface receptors extends beyond the single identified antigen, with substantial trafficking dysregulation in presumably well-segregated membrane proteins. One may propose that pathogenic mechanisms involved in autoimmune encephalitic syndromes should not be considered as single channelopathies, whereby autoantibodies solely impacting their isolated antigens, but instead as disease processes with wider and sometime convergent routes to pathogenesis. Considering the array of research that has focussed on single protein-mediated defects, for instance in genetic models of epilepsy or neurodevelopmental condition, we would suggest that the high degree of orchestrated co-regulation between many neuronal proteins, can itself become a substrate for pathogenesis.

# Methods

## Reagents and tools

See Table 1.

**Table 1.  Reagents and tools.**

| Reagent/resource | Reference or source | Identifier or catalogue number |
|---|---|---|
| **Antibodies** | | |
| Human anti-NMDAR IgG | Harald Pruss | #003–102 |
| Human anti-NMDAR IgG | Harald Pruss | #008–218 |
| Human anti-GABAaR IgG | Harald Pruss | #113–115 |
| Human anti-GABAaR IgG | Harald Pruss | #113–175 |
| Human Control IgG | Harald Pruss | #mGo-53 |
| Rabbit anti-GFP | Abcam | Cat# ab290 |
| Goat anti-rabbit Qdot-655 | Invitrogen | Cat# Q11422MP |
| Rabbit anti-flag | Sigma Aldrich | Cat# F2555 |
| Rabbit anti-GFP | Thermo Fischer | Cat# A6455 |
| Goat anti-rabbit AlexaFluor-488 | Invitrogen | Cat# A11008 |
| Donkey anti-rabbit AlexaFluor 568 | Invitrogen | Cat# A10042 |
| Mouse anti-gephyrin | Synaptic Systems | Cat# 147 111 |
| Mouse anti-gephyrin (Phospho S270) | Synaptic Systems | Cat# 147 011 |
| **Biological samples** | | |
| Embryonic primary hippocampal cultures | Created in lab | NA |
| **Recombinant DNA** | | |
| GluN1-SEP | Created in lab | NA |
| GluN1-Flag | Created in lab | NA |
| Gamma2 SEP | Created in lab | NA |
| GluA1-SEP | Created in lab | NA |
| Homer1C-dsRed | Created in lab | NA |
| Gephyrin-mVenus | Created in lab | NA |
| D1R (WT) | Created in lab | NA |
| D1R delta t2 | Created in lab | NA |
| **Bacterial and virus strains** | | |
| AAV-mDlx-GFP | | AddGene #83900 |
| AAV-CaMKII-GCaMP6f | James M. Wilson | AddGene #100834 |
| **Chemicals, peptides and recombinant proteins** | | |
| Phosphate buffered saline | Euromedex | Cat# ET330 |
| Bovine serum albumin | Sigma Aldrich | Cat# A9647 |
| Neurobasal | Thermo Fisher | Cat# 21103049 |
| B-27 supplement | Thermo Fisher | Cat# 17504044 |
| HBSS | Thermo Fisher | Cat# 14175095 |
| Glucose | Sigma Aldrich | Cat# G8270 |

**Table 1.** (continued)

| Reagent/resource | Reference or source | Identifier or catalogue number |
|---|---|---|
| GlutaMAX | Thermo Fisher | Cat# 35050087 |
| Calcium chloride | Sigma Aldrich | Cat# C8106 |
| Bicuculline methochloride | Tocris Bioscience | Cat# 1031 |
| D-APV | Tocris Bioscience | Cat# 0106 |
| Potassium chloride | Sigma Aldrich | Cat# P9541 |
| Potassium dihydrogen phosphate | Sigma Aldrich | Cat# 1.04871 |
| Magnesium chloride | Sigma Aldrich | Cat# M8266 |
| Magnesium sulphate | Sigma Aldrich | Cat# M2643 |
| Sodium chloride | Sigma Aldrich | Cat# S9888 |
| Sodium phosphate dibasic | Sigma Aldrich | Cat# S9763 |
| TTX | Tocris Bioscience | Cat# 1069 |
| Sucrose | Sigma Aldrich | Cat# S0389 |
| Sodium bicarbonate | Sigma Aldrich | Cat# S6014 |
| HEPES | Sigma Aldrich | Cat# H4034 |
| Caesium methanesulphonate | Sigma Aldrich | Cat# C1426 |
| EGTA | Sigma Aldrich | Cat# E3889 |
| QX-314 chloride | Tocris Bioscience | Cat# 2313 |
| Caesium hydroxide | Sigma Aldrich | Cat# 232041 |
| Magnesium-ATP | Sigma Aldrich | Cat# A9187 |
| Sodium-GTP | Sigma Aldrich | Cat# G8877 |
| **Experimental models: organisms/strains** | | |
| Sprague-Dawley rats | Janvier Labs | NA |
| **Software and algorithms** | | |
| MetaMorph | Molecular Devices | https://www.molecular devices.com |
| ImageJ (FIJI) | | https://imagej.nih.gov/ij |
| pClamp | Molecular Devices | https://www.molecular devices.com |
| Clampfit | Molecular Devices | https://www.molecular devices.com |
| GraphPad Prism | GraphPad | https://www.graphpad.com |

# Methods and protocols

### Derivation and synthesis of human monoclonal antibodies

Derivation of human monoclonal antibodies was performed by isolation of B-cells from CSF of patients with autoimmune encephalitis, or the peripheral blood of a healthy donor. Antibody-coding genes, of both heavy and light chains of G-isotype immunoglobulins were purified from B-cell lysates and amplified by PCR, before insertion into bacterial plasmids. For the recombinant expression of monoclonal antibodies, paired heavy and light chain-coding plasmids were co-transfected into HEK293T

cells and purified from cell culture supernatants, as previously described. In this study, human monoclonal antibodies #003–102 and 008–218, which binds to the GluN1 subunit of the NMDAR, was derived from patient with NMDAR encephalitis (antibody herein referred to as NMDAR mAb) and human monoclonal antibody #113–115 and 113–175, reactive to the α1β3-containing GABAaR, was derived from a patient with GABAaR encephalitis (antibody herein referred to as GABAaR mAb). Isotype-matched control antibody, mGo-53, was derived from blood of a healthy donor (antibody herein referred to as Control mAb). All human monoclonal antibodies used in this study were generated by the laboratory of Harald Prüss at Charité-Universitätsmedizin, Berlin, with all required informed consent from patients (Kreye et al, 2016; Kreye et al, 2021).

### Primary hippocampal cell cultures

Hippocampal cultures were prepared from embryonic day 18 Sprague-Dawley rat pups. Animal procedures were conducted in accordance with the European Community guidelines (Directive 2010/63/EU) regulating animal research, and were approved by the local Bordeaux Ethics Committee (APAFIS#3420-2015112610591204). Briefly, hippocampi were dissected in ice-cold HBSS containing Penicillin-Streptomycin (PS) and HEPES. Hippocampi were incubated with trypsin-EDTA and dissociated by mechanical trituration. Cell suspension—containing neurons and glia—was diluted in 60-mm sterile Petri dishes containing pre-warmed Neurobasal culture medium supplemented with horse serum and poly-L-lysine coated 18-mm coverslips, at a density of $250–275 \times 10^3$ cells per ml. Dishes were maintained at 37 °C in 5% $CO_2$ in a humidity-controlled incubator. For standard primary cultures, at 3 days in vitro (DIV), a full media exchange with serum-free Neurobasal/B-27 culture media was performed. Full media exchanges continued twice weekly until use. Coverslips were flipped onto astrocyte feeder layers, 3 h after plating, and maintained in this inverted configuration. At DIV 3, a full media exchange with serum-free Neurobasal/B-27 culture media, containing 5 μM cytosine arabinoside was performed to prevent astrocyte proliferation. Kaech and Banker (2006) protocols were used for experiments in which astrocytic expression of target surface proteins could interfere with data collection or add extraneous noise to imaging of neuronal cells (Kaech and Banker, 2006).

Where indicated, exogenous transgenes were introduced to neuronal cultures between DIV7-8 with a calcium phosphate transfection protocol. Briefly, a TE buffer solution containing purified bacterial plasmid DNA and 0.2 M $CaCl_2$ was added dropwise to an equal volume HEPES-based phosphate buffer to form fine plasmid-containing calcium phosphate precipitates. The mass of DNA used for all transfection conditions did not exceed 2 μg total DNA per coverslip. Depending on the experimental requirements, combinations of the following bacterial plasmids were used: pRcCMV-flag-rGrin1, pRcCMVaa-SEP-rGrin1, pRK5CMV-rGabrg2-SEP, pRK5CMV-HA-SEP-rGria1, pcDNA3.1CMV-dsRed-rHomer1c, pEGFP-N1-CMV-mVenus-rGphn, pcDNA3CMVHA-hDRD1(wt) and pcDNA3CMV-CFP-hDRD1(ΔT2). Coverslips were transferred to 12-well culture plates, containing pre-warmed Neurobasal culture medium supplemented with kynurenic acid, 50 μl of plasmid precipitate suspension was added to each well and incubated for 90 min. The remaining precipitate suspension was then washed, and coverslips were

returned to culture dishes until imaging at DIV 12–14. Auto-antibodies and/or pharmacological reagents were added to neuronal cultures by dilution into the cell culture media, during a full media exchange the day prior to imaging. High-affinity autoantibody samples (003–102 and 113–115) were diluted to a final concentration of 0.5 μg per ml of media; lower-affinity autoantibodies (008–218 and 113–175 were applied to a final concentration of 1.0 μg per ml. Where indicated, tetrodotoxin was applied to a final concentration of 20 nM, D-APV at 1 μM, bicuculline at 5 μM and casein kinase 1 (CK1) inhibitor, CKI7, at 100 μM.

### Organotypic hippocampal slice culture

Hippocampal slice cultures were prepared from Sprague-Dawley rat pups at postnatal day 5. Hippocampi were dissected in ice-cold dissection solution composed of 0.5 mM $CaCl_2$, 2.5 mM KCl, 0.7 mM $KH_2PO_4$, 2 mM $MgCl_2$, 0.3 mM $MgSO_4$, 50 mM NaCl, 0.9 mM $Na_2HPO_4$, 25 mM glucose, 2.7 mM $NaHCO_3$, 175 mM sucrose and 2 mM HEPES (pH 7.4, 320 mOsm). After isolation, 350-μm thick transverse hippocampal slices were prepared using a McIlwain tissue chopper. Slices were rested in dissection solution at 4 °C for 30 min, before plating onto PTFE membrane sections (FHLC01300, Millipore, UK) placed on Millicell cell culture inserts (PICM03050, Millipore, UK) and cultured in six-well plates containing pre-warmed slice culture media. Slice culture media was composed of 50% BME, 25% HBSS, 25% horse serum and supplemented with 25 mM glucose and GlutaMAX™ supplement. Slice cultures were maintained at 35 °C with 5% $CO_2$ in a humidity-controlled incubator. Full culture medium exchanges were performed the day after dissection, and then three times per week with pre-warmed slice culture medium, until electrophysiological recordings between DIV 12 and DIV 15. Where indicated, slice cultures were transfected using an adeno-associated viral strategy, to label inhibitory interneurons or to express a calcium reporter, GCaMPVI in excitatory neurons. Viral particles containing the plasmid *pAAV-mDlx-GFP* (purchased from AddGene, Plasmid #83900), or *pAAVCaMKII-GCaMPVI*, were applied directly onto hippocampal slices at DIV 1, during medium exchange. Hippocampal slices were exposed to viral particles until the subsequent medium exchange at DIV 4. Expression of soluble GFP was confirmed by immunofluorescence of sparsely localised cell bodies across hippocampal layers, between DIV 12 and DIV 15. In all cases, autoantibodies were applied to slice cultures the day before electrophysiological recordings, by dilution into the slice culture media. A 20 μl droplet of antibody-containing media was also applied directly onto the slice, above the membrane insert, to facilitate diffusion of autoantibodies into the cultured tissue.

### Quantum dot single-particle tracking

Hippocampal primary neurons transfected with either GluN1-SEP or GABAaR-SEP and Homer1c-DsRed or Gephyrin-mCherry as a synaptic marker respectively, were incubated for 10 min at 37 °C with rabbit polyclonal antibodies against GFP subunit (1/10,000). Neurons were then washed and incubated for 10 min at 37 °C with quantum dots 655 goat F(ab)2 anti-rabbit (Invitrogen, 1/10,000). Quantum dots were detected by using a mercury lamp and appropriate excitation/emission filters. Images were obtained with an acquisition time of 50 ms with up to 1000 consecutive frames. Signals were detected using an EM-CCD camera (Quantem, Roper

Scientific). Quantum dots were followed on randomly selected dendritic regions for up to 20 min. Quantum dot recording sessions were processed with the MetaMorph software (Universal Imaging Corp.). The instantaneous diffusion coefficient, D, was calculated for each trajectory, from linear fits of the first four points of the mean square-displacement versus time function using MSD(t) = 5r 2 4(t)=4Dt. The 2D trajectories of single molecules in the plane of focus were constructed by correlation analysis between consecutive images using a Vogel algorithm. The synaptic diffusion coefficient was calculated from GluN2-QD trajectories that were only present inside the synaptic area, defined by Gephyrin or Homer staining. The instantaneous diffusion coefficient is reported as the median 25–75% (interquartile range, IQR).

### Immunocytochemistry, imaging and analysis

Surface transfected GluN1-Flag, GluA1-SEP or γ2-GABAaR-SEP were specifically stained in live neurons using rabbit polyclonal antibodies against Flag (1/200, Sigma Aldrich, F2555, 15 min 37 °C) or against GFP (1/500, Thermo Fischer, A6455, 15 min 37 °C). Neurons were then fixed with 4% paraformaldehyde for 15 min. After blocking with 1% BSA for 1 h, neurons were incubated with secondary antibody Goat anti-rabbit Alexa 488 antibodies (Invitrogen, A11008, 1/1000, 1 h) for GluN1-Flag and GluA1-SEP or Donkey anti-rabbit Alexa 568 antibodies (Invitrogen, A10042, 1/1000, 1 h) for GABAaR-SEP. Neurons were washed, mounted and preparations were kept at 4 °C until observation using a spinning disk microscope Nikon Ni-E with spinning Yokogawa X1. Activation of astrocytes and microglial cells was investigated by immunostaining for GFAP or Iba1, respectively. Following incubation of organotypic hippocampal cultures with either NMDAR mAb, or antibody-free culture media, slices were briefly washed before fixation in 4% paraformaldehyde 4% sucrose PBS solution for 30 min at room temperature. Following fixation, slices were washed and aldehydic fluorescence was quenched with 50 mM ammonium chloride for 30 min. Slices were then washed again and submerged in permeabilization solution (0.5% Triton X-100 in PBS) and maintained overnight at 4 °C. Slices were washed before blocking with a 20% BSA PBS solution at 4 °C until the following day. Slices were then immunostained by submersion in 5% BSA PBS solution containing either rabbit antiGFAP (Abcam AB278054) or rabbit anti-Iba1 (WAKO 019-19741) at 1:1000 at 4 °C until the following day. Slices were then washed in 5% BSA PBS solution 4 times, over 2 h, at room temperature and under agitation. Secondary antibody (goat anti-rabbit alexafluor-488, Thermo Fisher A11008) was then added to 5% BSA PBS solution and slices were incubated at room temperature for 4 h, with agitation. Finally, slices were washed four times in PBS, over 2 h, before mounting in Vectashield anti-fade mounting medium, containing DAPI as a counterstain. A 200 μm spacer was added between the slide and cover glass to prevent compression and deformation of slices. Slices were kept at 4 °C until imaging. To quantify the level of GFAP and Iba1 expression and microglial and astrocytic cell morphology in organotypic slices, 20-μm thick zstacks were taken from the CA1 hippocampal subfield and maximum intensity projections were generated. A uniform threshold was applied across all projection images to identify the relative ratio of Iba1+ or GFAP+ pixels per stack projection, with the expectation that an increased activation of microglia or astrocytes will result in an increased expression of these intracellular proteins,

and thus an increased positive-pixel ratio. Immunostainings of endogenous gephyrin were carried out with the following protocols: Following 24-h incubation with experimental autoantibodies, hippocampal cultures were washed, fixed and quenched previously as described. Cultures were then permeabilized in 0.1% Triton X-100 PBS solution for 15 min at room temperature. Coverslips were then washed in PBS and blocked as described before, and subsequently incubated with either mouse anti-gephyrin (Synaptic Systems 147 111), at 1:1000 for 2 h at room temperature, to label gephyrin scaffolds independent of phosphorylation status; or alternatively with phospho-specific mouse anti-gephyrin (Synaptic Systems 147 011), under the same protocols, to label gephyrin scaffolds phosphorylated at S270. Coverslips were then washed and labelled with goat anti-mouse alexafluor-488 (Thermo Fisher A11001), for 1 h at room temperature before washing and mounting and storage as described above.

### Functional kinase assay

Kinase activity profiling was analysed by using a microarray assay containing 144 (STK) or 196 (PTK) phosphosites immobilised on a porous ceramic membrane, each of these phosphosites encoded in 13 amino acid long peptides which derived from literature or computational predictions with the phosphorylation of these phosphosites is then used to predict one or multiple upstream kinases (Protein tyrosine kinases for the PTK PamChip® and Serine threonine kinases for the STK PamChip®). In brief, STK or PTK microarrays were exposed to lysated neurones treated with NMDAR, GABAaR or control mAb for 24 h, following the manufacturer's protocol. Fluorescently labelled anti-phospho-antibodies were used to detect the phosphorylation activity of kinases present in the sample. Images of each array were taken at several exposure times by a camera in the Pamgene workstation. Images are later used by the BioNavigator® software for image quantification, quality control, statistical analysis, visualisation and interpretation.

### Electrophysiology

Spontaneous excitatory and inhibitory postsynaptic currents (sEPSC and sIPSC, respectively) were recorded using a whole-cell patch-clamp technique in voltage-clamp configuration. In summary, hippocampal slice cultures were transferred to the chamber of an upright microscope, containing 34 °C carbogen-bubbled extracellular solution composed of 126 mM NaCl, 3.5 mM KCl, 2 mM $CaCl_2$, 1.3 mM $MgCl_2$, 1.2 mM $NaH_2PO_4$, 25 mM $NaHCO_3$ and 12.1 mM glucose (pH 7.4, 310 mOsm), perfused at approximately 2 ml per min. Neurons in the pyramidal cell layer of the CA1 hippocampal subfield were visualised using infrared differential interference contrast imaging. Patch pipettes for voltage-clamp recordings were filled with intracellular solution composed of 130 mM CsMeS, 20 mM HEPES, 0.2 mM EGTA, 5 mM QX-314·Cl, 2 mM NaCl, 4 mM Mg-ATP and 0.4 mM Na-GTP (pH 7.3, 290 mOsm). Pipette resistance ranged from 4–6 MΩ. After achieving a whole-cell configuration, AMPA receptor-mediated sEPSCs were recorded while clamping the membrane potential at −70 mV. After 10 min of sEPSC recording, cells were then clamped at 0 mV to record 10 min of GABAa receptor-mediated sIPSCs. Traces were recorded using a Multiclamp 700B amplifier and a Digidata 1550B interface controlled by Clampex 10.7 (Molecular Devices). Series resistance was monitored throughout the

experiment by a brief voltage step of −5 mV at 30 s intervals. Data were discarded if series resistance was found to vary by more than 20% during the recording session. Traces were analysed using Clampfit software, in which synaptic events were detecting using an automated template search protocol. To permit correct characterisation of synaptic event amplitudes and kinetic parameters, extracted events were manually inspected for adherence to inclusion criteria. Specifically, events were discarded in instances where two synaptic events were found to be superimposed, where events occurred during a series resistance test or when peak amplitude failed to exceed 2 standard deviations of the baseline. Amplitudes of all detected events are displayed as cumulative frequency distributions, and for quantification, mean event amplitudes were calculated for each cell recorded (Control mAb: $n = 25$ cells; NMDAR mAb: $n = 9$ cells & GABAaR mAb: $n = 15$ cells). Error bars represent the standard error of the mean.

Quantification of tonic GABAergic currents was performed by the same recording configuration described above for GABAa receptor-mediated inhibitory currents, with the modification that tetrodotoxin was added to the extracellular recording solution, to a final concentration of 2 μM. After 5 min recording of a stable baseline period, bicuculline was washed into the recording solution, at a concentration of 5 μM. The amplitude of tonic GABAergic currents was estimated by recording the mean baseline before addition of bicuculline and subtracting the measured baseline after addition and 5-min wash-in of the drug. Control Experiments without BCC wash-in were used to ensure that baselines did not significantly drift during a 15-min recording period, by generating a "predicted" baseline based on linear regression fitting of the first 5 min of recording and comparing to the experimentally measured value (Fig. EV2I).

Characterisation of intrinsic cellular excitability of hippocampal pyramidal cells was performed using a whole-cell current-clamp approach. In these experiments, patch pipettes were filled with intracellular solution composed of 127 mM K-gluconate, 8 mM KCl, 10 mM HEPES, 15 mM phosphocreatine, 4 mM Mg-ATP and 0.3 mM Na-GTP. The recording paradigm consisted of 12 sweeps of 2 s duration. A 500 ms variable-amplitude current step was introduced between 250 and 750 ms on each sweep, outside the current step epoch, cells were injected with 0 pA of current. Current step amplitudes ran sequentially from hyperpolarising (−150 pA) to depolarising (400 pA), in 50 pA increments. Traces were recorded using a Multiclamp 700B amplifier and a Digidata 1550B interface controlled by Clampex 10.7 (Molecular Devices), and data were analysed using Clampfit. Input–output curves were generated by extracting the number of action potential discharges inside each applied depolarising current step. Action potentials were identified using a template search algorithm in Clampfit. Resting membrane potential was estimated in each recorded cell by measuring the mean membrane potential during the final 1 s of each sweep, such that measurement would not be affected by variable-amplitude current epochs. The threshold for action potential firing was calculated using the first observed action potential in each cell recording. The threshold was defined as the lowest mV value at which the action potential waveform exceeded 20 mV per millisecond. Rheobase was defined as the lowest amplitude of an applied current step, in which an action potential was elicited. Input resistance was calculated from each of the three hyperpolarising current steps (−150, −100, and −50 pA), using the

equation (Input Resistance) = (Mean Baseline Voltage)/(Injected Current). Intrinsic properties of inhibitory interneurons were characterised as described above, in hippocampal slices infected with an AAV-inducing soluble cytoplasmic GFP expression under an interneuron-specific (mDlx) promoter (see slice culture methods above), where patched cells were identified as GFP-positive under 488 nm LED illumination. Spontaneous action potential firing frequencies were quantified using a cell-attached voltage-clamp configuration. For these experiments, patch pipettes were filled with 150 mM NaCl. After formation of a seal between the cell membrane and patch pipette, in the order of gigaohm resistance, the voltage clamp was set to a potential at which zero current was injected through the pipette. Traces were recorded using a Multiclamp 700B amplifier and a Digidata 1550B interface controlled by Clampex 10.7 (Molecular Devices), and data were analysed using Clampfit. Action potential spikes were extracted from trances using a template search algorithm.

### Calcium imaging

Organotypic hippocampal cultures were virally transfected using an AAV, inducing the expression of calcium reporter, GCaMPVI, under the excitatory neuron-specific promotor for CaMKII. Transfection was performed as described above at DIV 1 (see organotypic hippocampal slice culture methods) after 24 h exposure to either Control, NMDAR or GABAaR mAb, the CA1 subfield of hippocampal slice cultures was imaged using a spinning disk confocal microscope (Nikon Ni-E with spinning Yokogawa X1), at a frame rate of approximately 3 Hz. Slice cultures were maintained in warmed (37 °C) BrainPhys neuronal culture media throughout the imaging procedure, where time-lapse recordings in a single Z-plane were acquired for 7 min. Acquired image stacks were then analysed using a set of custom-built scripts for ImageJ software. Briefly, somatic regions of interest (ROIs) were selected by creating a "virtual δF/F image stack". Mean fluorescence intensity was then measured in each ROI across the 7 min recording interval, and fluorescence fluctuations were determined by creating δF/F traces, where δF = Mean fluorescence intensity at frame $n$; and F = mean fluorescence intensity of the prior five consecutive frames ($n - 6$ to $n - 1$ frames). Spikes were then identified by running a spike detection algorithm, which searched the δF/F trace for incidents where point values on the δF/F function exceeds three times the standard deviation of the full δF/F trace. Spike detections were manually inspected in a subset of ROIs in each recording to ensure adequate and equivalent detections across acquisitions.

### Quantification and statistical analysis

Comparisons between groups were either performed with parametric statistical tests (Student's $t$ test, one-way ANOVA followed by post hoc test) or with non-parametric Mann–Whitney test (single quantum dots). Statistical tests and the number of samples are described in the figure legends. Significance levels were defined as *$P \leq 0.05$, **$P \leq 0.01$, ***$P \leq 0.001$, ****$P \leq 0.0001$.

## Data availability

The datasets produced in this study are available in the following databases: BioImage Archive, Accession number S-BSST1238.

## Peer review information

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

## Acknowledgements

We would like to thank the Cell Biology Facility, especially Delphine Bouchet, Constance Manso, Emeline Verdier and Natacha Retailleau, and lab members for constructive discussions. We also thank the Bordeaux Imaging Center, a service unit of the CNRS-INSERM and Bordeaux University, a member of the national infrastructure France BioImaging (ANR-10-INBS-04-01). This work was supported by the Centre National de la Recherche Scientifique (LG), Agence Nationale de la Recherche (ANR PRC DopamineHub to LG), European Union's Horizon 2020 research and innovation programme under the Marie Sklodowska-Curie grant agreement No. 813986 (H2020-MSCA-ITN Syn2Psy; DH and LG), EraNet Neuron Mental Disorders Program (Project Autoscale; to LG and 01EW1901 to CG), the European Research Council Synergy grant (ENSEMBLE, #951294 to LG), Fondation pour la Recherche Médicale (SPF201909009269 to MP), the German Research Foundation (DFG, GE2519/8-1, GE2519/9-1, GE2519/11-1 to CG), the Interdisziplinäres Zentrum für Klinische Forschung (IZKF) Jena, the Foundation Else Kröner-Fresenius-Stiftung within the Else Kröner Research School for Physicians "AntiAge" (MC), GPR BRAINUniversité de Bordeaux (LG), Académie Nationale de Médecine (LG), the German Research Foundation (DFG; grants FOR3004, PR1274/3-1, PR1274/5-1, and PR1274/9-1 to HP), the Helmholtz Association (HIL-A03 BaoBab) and the German Federal Ministry of Education and Research (Connect-Generate 01GM1908D; to HP).

## Author contributions

**Daniel Hunter**: Conceptualisation; Data curation; Formal analysis; Validation; Investigation. **Mar Petit-Pedrol**: Conceptualisation; Data curation; Formal analysis; Validation; Investigation; Visualisation; Writing—original draft; Writing—review and editing. **Dominique Fernandes**: Data curation; Formal analysis; Validation; Investigation; Visualisation. **Nathan Bénac**: Data curation; Formal analysis; Investigation; Visualisation. **Catarina Rodrigues**: Data curation; Validation; Investigation. **Jakob Kreye**: Resources. **Mihai Ceanga**: Resources. **Harald Prüss**: Resources. **Christian Geis**: Resources. **Laurent Groc**: Conceptualisation; Data curation; Supervision; Funding acquisition; Visualisation; Writing—original draft; Project administration; Writing—review and editing.

## Disclosure and competing interests statement

The authors declare no competing interests.

# Expanded View Figure

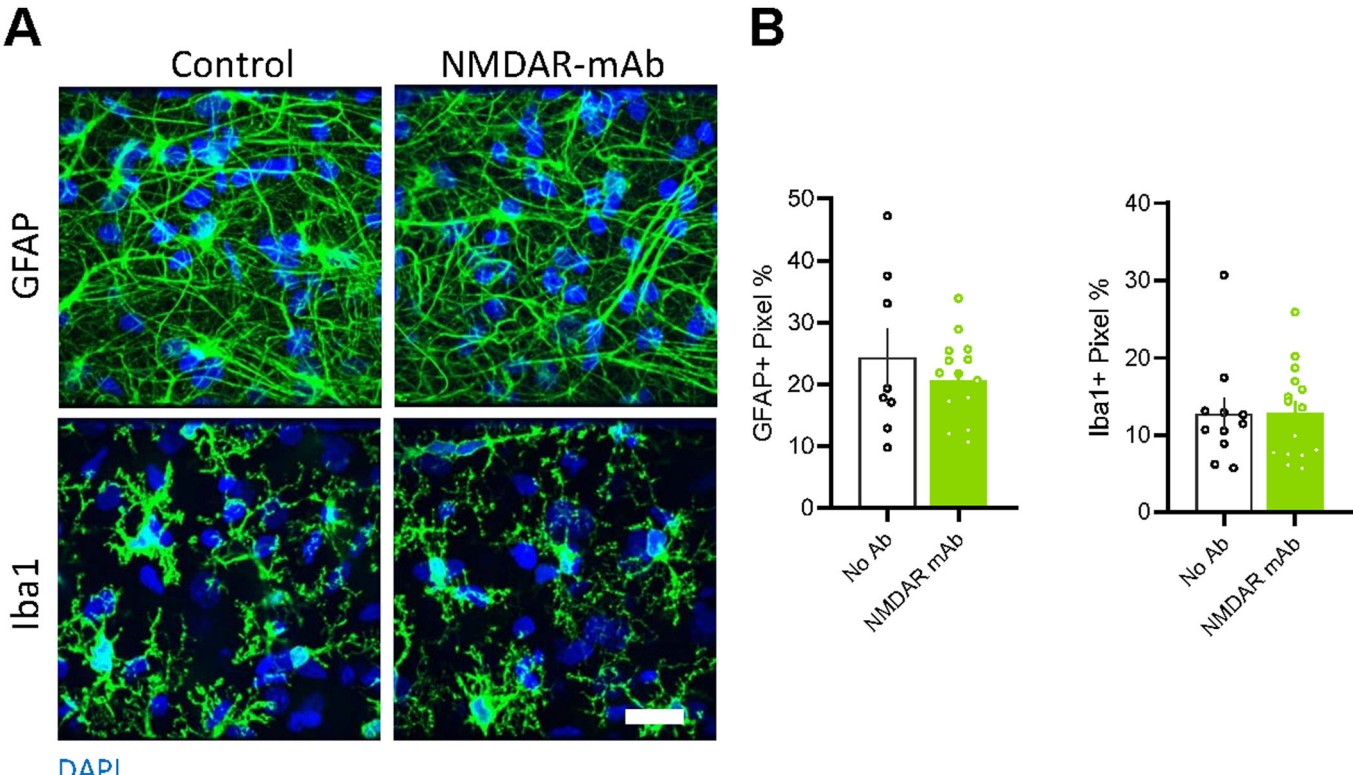

**Figure EV1.  NMDAR mAb do not alter glial and microglial coverage.**

(A) Immunocytochemical staining of GFAP-positive and Iba1-positive cells (green), with a DAPI counterstaining (blue), in hippocampal slices exposed to buffer (control) or NMDAR mAb. Scale bar = 20 μm. (B) Quantification and comparison of the GFAP and Iba1 fluorescence (fraction of positive pixel) between control and NMDAR mAb conditions (No Ab, $n = 9$; NMDAR mAb, $n = 15$; $P > 0.05$ for all stainings). Data information: All error bars represent the standard error of the mean.

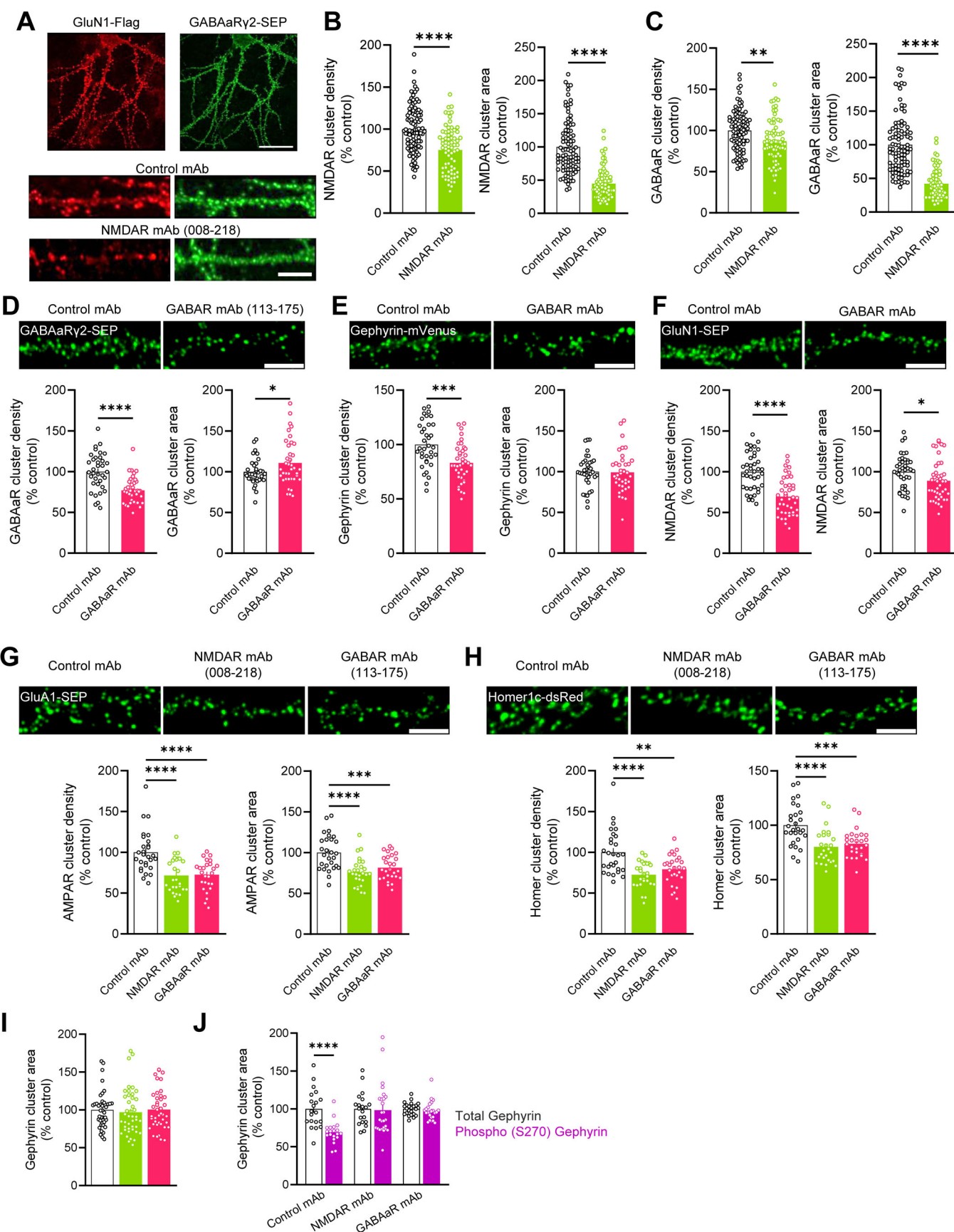

◀ **Figure EV2. Monoclonal antibody impacts of macro-organisational cluster area.**

(**A**) Immunocytochemical staining of surface NMDAR (GluN1-Flag, red) or GABAaR (Gamma2 SEP, green) in neurons exposed to control mAb or NMDAR mAb (clone 008–218). Scale bars = 20 µm (upper panels), 2 µm lower panels. (**B, C**) Mean NMDAR (**B**) or GABAaR (**C**) cluster density and area, normalised to the Control mAb condition (Control mAb $n = 90$ cells; NMDAR mAb $n = 76$; Student $t$ test). (**D–F**) Mean synaptic GABAaR (**D**—Control mAb $n = 36$ cells; GABAR mAb $n = 39$ cells), gephyrin (**E**—Control mAb $n = 36$ cells; GABAR mAb $n = 39$ cells), and NMDAR (**F**—Control mAb $n = 40$ cells; GABAR mAb $n = 42$ cells) cluster density and area in neurons exposed to GABAaR (clone 113–175) mAb (Student $t$ test). Scale bars = 2 µm. (**G, H**) Mean synaptic AMPAR (**G**) and Homer1c (**H**) cluster density and area in neurons exposed to Control ($n = 28$ cells), NMDAR (clone 008–218, $n = 27$ cells), or GABAaR (clone 113–175, $n = 27$ cells) mAb (one-way ANOVA). Scale bars = 2 µm. (**I**) Mean gephyrin cluster area, normalised to Control mAb condition (Control mAb $n = 40$ cells; NMDAR mAb $n = 43$ cells; GABAaR mAb $n = 41$ cells; One-way ANOVA). (**J**) Total versus phosphorylated gephyrin puncta density, normalised to the level of total gephyrin staining (Control mAb total gephyrin $n = 20$ cells; Control mAb Phospho-Gephyrin $n = 20$ cells; NMDAR mAb total gephyrin $N = 20$ cells; NMDAR mAb Phospho-Gephyrin $n = 26$ cells; GABAaR mAb total gephyrin $n = 24$ cells; GABAaR mAb Phospho-Gephyrin $n = 24$ cells; Multiple $t$ tests with Benjamini and Hochberg correction for false discovery rate). Data information: All error bars represent the standard error of the mean. Significance levels are represented as $*P < 0.05$, $**P < 0.01$, $***P < 0.001$ and $****P < 0.0001$.

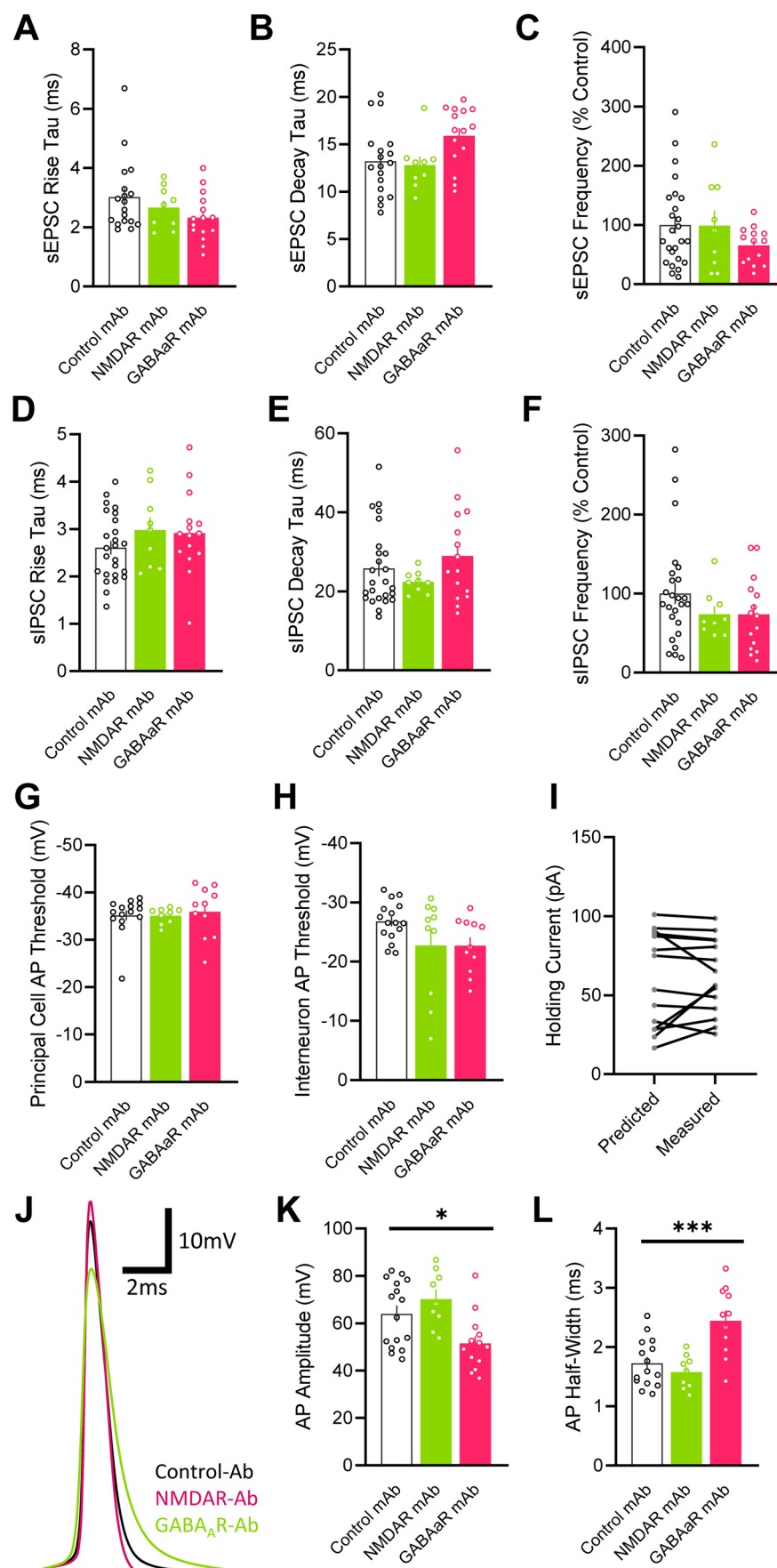

**Figure EV3. Autoantibodies do not alter synaptic current kinetics or action potential thresholds.**

(A–C) Mean sEPSC rise and decay tau, and frequency after exposure to control, NMDAR or GABAaR mAb for 24 h (Control mAb $n = 18$ cells; NMDAR mAb $n = 9$ cells; GABAaR mAb $n = 15$ cells; one-way ANOVA). (D–F) Mean sIPSC rise and decay tau and frequency after exposure to control, NMDAR or GABAaR mAb for 24 h (Control mAb $n = 25$ cells; NMDAR mAb $n = 9$ cells; GABAaR mAb $n = 15$ cells; one-way ANOVA). (G, H) Action potential threshold in CA1 hippocampal principal cells (Control mAb $n = 15$ cells; NMDAR mAb $n = 9$ cells; GABAaR mAb $n = 11$ cells; one-way ANOVA) and interneurons (Control mAb $n = 16$ cells; NMDAR mAb $n = 10$ cells; GABAaR mAb $n = 11$ cells; one-way ANOVA). (I) Predicted holding current of mIPSC baseline traces versus experimentally determined holding current ($n = 14$, Student's paired $t$ test). (J) Representative action potential traces, collected from CA1 pyramidal cells in current-clamp. (K, L) Mean action potential amplitude and half-width (Control mAb $n = 16$ cells; NMDAR mAb $n = 19$ cells; GABAaR mAb $n = 14$ cells; one-way ANOVA). Data information: All error bars represent the standard error of the mean. Significance levels are represented as *$P < 0.05$, **$P < 0.01$, ***$P < 0.001$ and ****$P < 0.0001$.

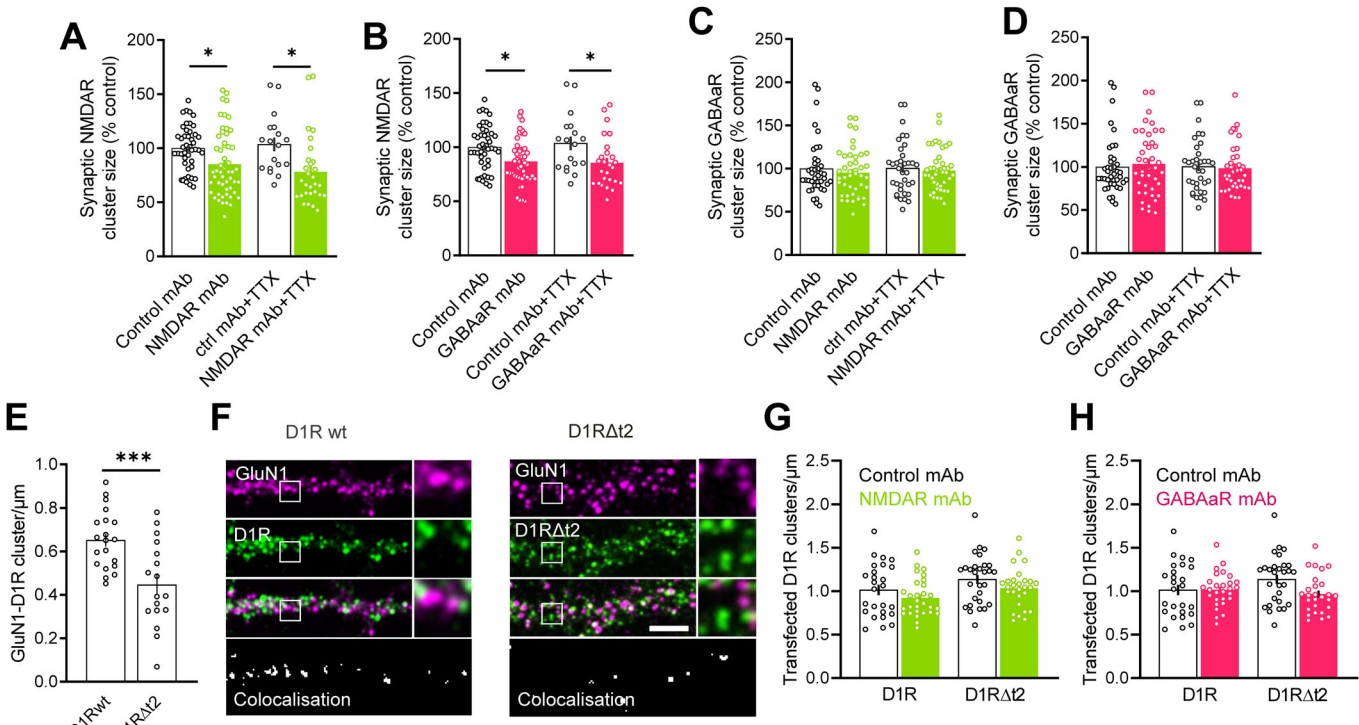

**Figure EV4. Excitatory and inhibitory synaptic crosstalk is activity-dependent and D1RΔt2 transfection reduced GluN1-D1R interaction without altering D1R organisation.**

(A) Cell quantification of synaptic NMDAR cluster density after 24 h exposure to Control mAb or NMDAR mAb, in the presence or absence of TTX (Control mAb $n = 46$ cells; NMDAR mAb $n = 55$; Control mAb +TTX $n = 18$; NMDAR mAb +TTX $n = 29$; one-way ANOVA). (B) Cell quantification of synaptic NMDAR cluster density after 24 h exposure to Control mAb or GABAaR mAb, in the presence or absence of TTX (Control mAb $n = 46$ cells; GABAaR mAb $n = 46$; Control mAb +TTX $n = 18$; GABAaR mAb +TTX $N = 26$; one-way ANOVA). (C) Cell quantification of synaptic GABAaR cluster density after 24 h exposure to Control mAb or NMDAR mAb, in the presence or absence of TTX (Control mAb $n = 40$ cells; NMDAR mAb $n = 43$; Control mAb +TTX $n = 35$; NMDAR mAb +TTX $n = 43$; one-way ANOVA). (D) Cell quantification of synaptic GABAaR cluster density after 24 h exposure to Control mAb or GABAaR mAb, in the presence or absence of TTX (Control mAb $n = 40$ cells; GABAaR mAb $n = 41$; Control mAb +TTX $n = 35$; GABAaR mAb +TTX $n = 37$; one-way ANOVA). (E) Mean thresholded GluN1-D1R colocalised cluster density after D1R-WT and D1R-Δt2 transfection (D1R-WT $n = 20$ cells; D1R-Δt2 $n = 17$ cells, Students $t$ test). (F) Representative immunostainings of D1R-WT and D1R-Δt2 transfected neuronal dendrites expressing GluN1 and D1Rs. Scale bar = 10 μm. (G) Mean D1R cluster density after transfection with D1R-WT and D1R-Δt2 constructs and exposure to Control or NMDAR mAb (D1R: Control mAb $n = 26$ cells; NMDAR mAb $n = 29$ cells; D1R-Δt2: Control mAb $n = 29$ cells; NMDAR mAb $n = 28$ cells; one-way ANOVA). (H) Mean D1R cluster density after transfection with D1R-WT and D1R-Δt2 constructs and exposure to Control or GABAaR mAb (D1R: Control mAb $n = 26$ cells; GABAaR mAb $n = 28$ cells; D1R-Δt2: Control mAb $n = 29$ cells; GABAaR mAb $n = 27$ cells; one-way ANOVA). Data information: All error bars represent the standard error of the mean. Significance levels are represented as *$P < 0.05$, **$P < 0.01$, ***$P < 0.001$ and ****$P < 0.0001$.

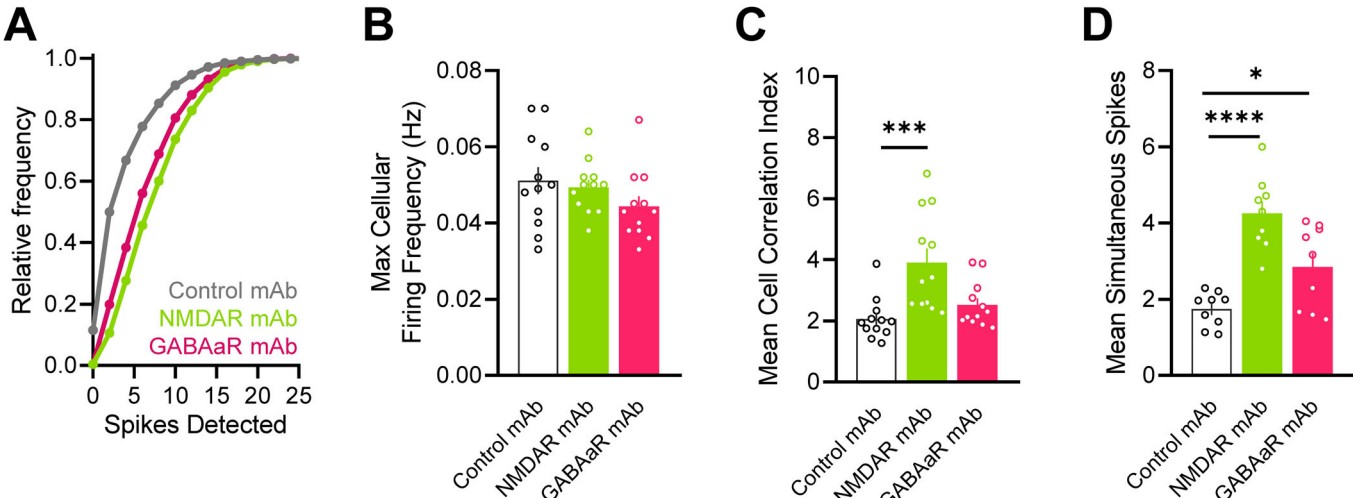

**Figure EV5. Autoantibody exposure impacts cell firing and network synchrony.**

(A) Cumulative frequency distributions of cellular somatic calcium transients from CA1 principal cells exposed to Control, NMDAR and GABAaR mAb (Control mAb $n = 1946$ spikes; NMDAR mAb $n = 2177$ spikes; GABAaR mAb $n = 2163$ spikes). (B) Maximum cellular firing frequencies from CA1 principal cells in recorded networks (Control mAb $N = 12$ slices; NMDAR mAb $n = 12$ slices; GABAaR mAb $n = 12$ slices; one-way ANOVA). (C, D) Mean cell correlation index and simultaneous spike rate for all CA1 principal cells across a recorded network (Control mAb $n = 12$ slices; NMDAR mAb $n = 12$ slices; GABAaR mAb $n = 12$ slices; one-way ANOVA). Data information: All error bars represent the standard error of the mean. Significance levels are represented as *$P < 0.05$, **$P < 0.01$, ***$P < 0.001$ and ****$P < 0.0001$.

