## [Peer Review File · EMBO Reports]

Converging synaptic and network dysfunctions in distinct autoimmune encephalitis

Mar Petit-Pedrol, Daniel Hunter, Dominique Fernandes, Nathan Benac, Catarina Rodrigues, Jakob Kreye, Mihai Ceanga, Harald Prüss, Christian Geis, and Laurent Groc

DOI: [10.15252/embr.202358073](https://doi.org/10.15252/embr.202358073)

Corresponding author(s): Laurent Groc (laurent.groc@u-bordeaux.fr)

Review Timeline:

Submission Date:	29th Aug 23
Editorial Decision:	10th Oct 23
Revision Received:	23rd Nov 23
Editorial Decision:	12th Dec 23
Revision Received:	18th Dec 23
Accepted:	2nd Jan 24

Transaction Report:

Dear Dr. Groc

Thank you for the submission of your research manuscript to our journal. We have now received the full set of referee reports that is copied below.

As you will see, the referees acknowledge that the findings are potentially interesting, but they also raise concerns that the conclusions rely on single antibodies targeting GABA A and NMDAR, respectively. I have discussed this concern further with the referees and referee 3 indicated that testing a second set of antibodies, if available, would be desirable. This further analysis would clarify whether the current NMDAR and GABA A antibody result in one of potentially several functional outcomes or whether the current effects could be considered to be at least not a completely unique response. Please address this and all other concerns in the manuscript, either by performing further experiments or in the discussion.

Given these constructive comments, we would like to invite you to revise your manuscript with the understanding that the referee concerns (as detailed above and in their reports) must be fully addressed and their suggestions taken on board. Please address all referee concerns in a complete point-by-point response. Acceptance of the manuscript will depend on a positive outcome of a second round of review. It is EMBO Reports policy to allow a single round of revision only and acceptance or rejection of the manuscript will therefore depend on the completeness of your responses included in the next, final version of the manuscript.

We realize that it is difficult to revise to a specific deadline. In the interest of protecting the conceptual advance provided by the work, we recommend a revision within 3 months (January 10th, 2024). Please discuss the revision progress ahead of this time with the editor if you require more time to complete the revisions.

I am also happy to discuss the revision further via e-mail or a video call, if you wish.

*******IMPORTANT NOTE:**

We perform an initial quality control of all revised manuscripts before re-review. Your manuscript will FAIL this control and the handling will be delayed IN CASE the following APPLIES:

- 1) A data availability section providing access to data deposited in public databases is missing. If you have not deposited any data, please add a sentence to the data availability section that explains that.
- 2) Your manuscript contains statistics and error bars based on $n=2$. Please use scatter blots in these cases. No statistics should be calculated if $n=2$.

When submitting your revised manuscript, please carefully review the instructions that follow below. Failure to include requested items will delay the evaluation of your revision. *****

- 1) a .docx formatted version of the manuscript text (including legends for main figures, EV figures and tables). Please make sure that the changes are highlighted to be clearly visible.
- 2) individual production quality figure files as .eps, .tif, .jpg (one file per figure). Please download our Figure Preparation Guidelines (figure preparation pdf) from our Author Guidelines pages <https://www.embopress.org/page/journal/14693178/authorguide> for more info on how to prepare your figures.
- 3) a .docx formatted letter INCLUDING the reviewers' reports and your detailed point-by-point responses to their comments. As part of the EMBO Press transparent editorial process, the point-by-point response is part of the Review Process File (RPF), which will be published alongside your paper.
- 4) a complete author checklist, which you can download from our author guidelines (<<https://www.embopress.org/page/journal/14693178/authorguide>>). Please insert information in the checklist that is also reflected in the manuscript. The completed author checklist will also be part of the RPF.
- 5) Please note that all corresponding authors are required to supply an ORCID ID for their name upon submission of a revised manuscript (<<https://orcid.org/>>). Please find instructions on how to link your ORCID ID to your account in our manuscript tracking system in our Author guidelines (<<https://www.embopress.org/page/journal/14693178/authorguide#authorshipguidelines>>)

6) We replaced Supplementary Information with Expanded View (EV) Figures and Tables that are collapsible/expandable online. A maximum of 5 EV Figures can be typeset. EV Figures should be cited as "Figure EV1, Figure EV2" etc... in the text and their respective legends should be included in the main text after the legends of regular figures.

7) Please note that a Data Availability section at the end of Materials and Methods is now mandatory. In case you have no data that requires deposition in a public database, please state so instead of refereeing to the database. See also <<https://www.embopress.org/page/journal/14693178/authorguide#dataavailability>>. Please note that the Data Availability Section is restricted to new primary data that are part of this study.

Additional information on source data and instruction on how to label the files are available <<https://www.embopress.org/page/journal/14693178/authorguide#sourcedata>>.

10) Figure legends and data quantification:

- the name of the statistical test used to generate error bars and P values,
- the number (n) of independent experiments (please specify technical or biological replicates) underlying each data point,
- the nature of the bars and error bars (s.d., s.e.m.)

- If the data are obtained from n {less than or equal to} 5, show the individual data points in addition to the SD or SEM.

- If the data are obtained from n {less than or equal to} 2, use scatter blots showing the individual data points.

11) Our journal encourages inclusion of *data citations in the reference list* to directly cite datasets that were re-used and obtained from public databases. Data citations in the article text are distinct from normal bibliographical citations and should directly link to the database records from which the data can be accessed. In the main text, data citations are formatted as follows: "Data ref: Smith et al, 2001" or "Data ref: NCBI Sequence Read Archive PRJNA342805, 2017". In the Reference list, data citations must be labeled with "[DATASET]". A data reference must provide the database name, accession number/identifiers and a resolvable link to the landing page from which the data can be accessed at the end of the reference. Further instructions are available at <<https://www.embopress.org/page/journal/14693178/authorguide#referencesformat>>.

12) All Materials and Methods need to be described in the main text. We would encourage you to use 'Structured Methods', our new Materials and Methods format. According to this format, the Materials and Methods section should include a Reagents and Tools Table (listing key reagents, experimental models, software and relevant equipment and including their sources and relevant identifiers) followed by a Methods and Protocols section in which we encourage the authors to describe their methods using a step-by-step protocol format with bullet points, to facilitate the adoption of the methodologies across labs. More information on how to adhere to this format as well as downloadable templates (.doc or .xls) for the Reagents and Tools Table can be found in our author guidelines: <

<https://www.embopress.org/page/journal/14693178/authorguide#manuscriptpreparation>>. An example of a Method paper with Structured Methods can be found here: <<https://www.embopress.org/doi/10.15252/msb.20178071>>.

13) As part of the EMBO publication's Transparent Editorial Process, EMBO Reports publishes online a Review Process File to accompany accepted manuscripts. This File will be published in conjunction with your paper and will include the referee reports, your point-by-point response and all pertinent correspondence relating to the manuscript.

Yours sincerely,

Referee #1:

Hunter et al. explored the idea that autoimmune encephalitis is associated with common neuropsychiatric symptoms and that patient-derived monoclonal autoantibodies can affect NMDA receptors and GABAergic synaptic transmission. This study is supported by big ideas, backed by robust experimental techniques. Although the experiments themselves are performed in animals, the clinical relevance of the results obtained is significant. Single-molecule tracking around synapses using multicolor quantum dots is a state-of-the-art technique that is unique to this laboratory, and in this sense, the results obtained are highly valuable and informative. I would recommend that this paper be published in EMBO Reports.

My only concern is that the authors only tried one type of antibody for each receptor. Would the effect on membrane diffusion of the receptors differ between the epitopes of antibodies? If different antibodies are available, they should be tried; if these experiments are difficult, I would recommend that the authors mention the possibility that different antibodies may have different effects and discuss the general scope of the implications (or the limitations) of this study.

Referee #2:

This is a very interesting manuscript addressing the biological basis for the confluence of symptomatology in distinct synaptic immunity disorders. The experiments are well performed and compelling, and the data are not over interpreted. I have only 2 comments the authors could expand on.

1. Given the role of dopaminergic input in the mechanisms proposed, would the present work extend to the dyskinesias associated with antiNMDAR encephalitis?
2. Calpain is an NMDAR activated protein in many cases. Could this also link to the control of gephyrin.

Referee #3:

Hunter et al describe effects of patients' autoantibodies against NMDAR and GABA A receptors on NMDAR, AMPAR and GABA A receptor distribution and function. Accordingly, incubation of HC at 14 DIV with either antibody increased NMDAR and decrease GABA A diffusion coefficients. Despite opposite effects on diffusion, the end result is for both, NMDAR and GABA A antibodies, a decrease in cluster density of NMDARs and GABA A. NMDAR but not GABA A cluster size and GluA1 cluster size but not density is also reduced by both antibodies.

These cell biological data are nicely complemented by electrophysiological analysis of spontaneous EPSCs. Accordingly, NMDAR and GABA A autoantibodies also decreased amplitudes of AMPAR-mediated sEPSC and, less so, GABA A -mediated

sIPSC so that the E/I balance is shifted towards excitation.

Mechanistically the authors identify several kinase activities that are affected by the two different autoantibodies. A common effect was an increase in GSK3 kinase activity. GSK3 interacts with casein kinase 1 (CK1), which regulates the interaction between the dopamine D1 receptor and NMDARs. The authors found that the CK1 inhibitor CK17 prevented the effect of the GABA A antibody on NMDARs. When they expressed the D1 receptor with a truncation of its NMDAR binding site, the effect of the NMDAR antibody on GABA A but not NMDAR density was lost. Similarly, the truncated D1 prevented the effect of GABA A antibodies on NMDAR but not GABA A. Accordingly, the cross-talk between NMDAR and GABA A binding antibodies with the opposite receptor seems to require the interaction between D1 and NMDARs when this was not required for antibody effects onto the targeted receptor.

Finally, the GABA A but not NDMAR antibody resulted in hyperexcitability of the HC.

Major concerns:

1. The authors describe only a single autoantibody against NMDAR and one against GABA A. It is unclear whether this is a specific effect of these two antibodies or applies more generally to other autoantibodies against these receptors.
2. Expression of D1 seems minimal if not absent in CA1/CA2/CA3 pyramidal neurons (but present in dentate gyrus) based on several reports. Could the related D5 receptor mediate some of the effects ascribed to ectopically expressed D1 receptor?

Reviewers' comments and responses

Reviewer #1

Hunter et al. explored the idea that autoimmune encephalitis is associated with common neuropsychiatric symptoms and that patient-derived monoclonal autoantibodies can affect NMDA receptors and GABAergic synaptic transmission. This study is supported by big ideas, backed by robust experimental techniques. Although the experiments themselves are performed in animals, the clinical relevance of the results obtained is significant. Single-molecule tracking around synapses using multicolor quantum dots is a state-of-the-art technique that is unique to this laboratory, and in this sense, the results obtained are highly valuable and informative. I would recommend that this paper be published in EMBO Reports.

Response: We greatly thank the reviewer for the positive comment and evaluation.

My only concern is that the authors only tried one type of antibody for each receptor. Would the effect on membrane diffusion of the receptors differ between the epitopes of antibodies? If different antibodies are available, they should be tried; if these experiments are difficult, I would recommend that the authors mention the possibility that different antibodies may have different effects and discuss the general scope of the implications (or the limitations) of this study.

Response: We agree that one antibody greatly thank the reviewer for the positive comments. In order to address this question, we have performed new series of experiments using two additional autoantibodies. We now demonstrate that different autoantibodies directed either against the NMDAR (clone 008-218; Ly et al., J. Neurol., 2018) or GABA_AR (clone 113-175; do not inhibit GABA_AR-mediated current; Kreye et al., JCI, 2021) produce similar effect on the glutamatergic and GABAergic receptor clusters that the ones presented in the initial study. Indeed, a 24h incubation with each of these new autoantibodies reduce the immunodetection of NMDAR and GABA_AR clusters. These data have now been added into the revised manuscript (Fig EV2).

Reviewer #2

This is a very interesting manuscript addressing the biological basis for the confluence of symptomatology in distinct synaptic immunity disorders. The experiments are well performed and compelling, and the data are not over interpreted. I have only 2 comments the authors could expand on.

- 1. Given the role of dopaminergic input in the mechanisms proposed, would the present work extend to the dyskinesias associated with antiNMDAR encephalitis?*
- 2. Calpain is an NMDAR activated protein in many cases. Could this also link to the control of gephyrin.*

Response: We greatly thank the reviewer for the positive comment and evaluation. The two interesting remarks have been taken into account in our revised manuscript. Regarding dyskinesia it is possible that the altered dopaminergic signaling induced by NMDAR mAb contribute to symptoms. Regarding the calpain, it is a likely candidate to mediate mAb-induced dismantling of gephyrin cluster. Future investigations will be needed to precisely define the role of calpain in this process.

Reviewer #3

We greatly thank the reviewer for his/her comment and critics.

Major concerns:

- 1. The authors describe only a single autoantibody against NMDAR and one against GABA A. It is unclear whether this is a specific effect of these two antibodies or applies more generally to other autoantibodies against these receptors.*

Response: We agree that one antibody greatly thank the reviewer for the positive comments. In order to address this question, we have performed new series of experiments using two additional autoantibodies. We now demonstrate that different autoantibodies directed either against the NMDAR clone 008-218; Ly et al., J.

Neurol., 2018) or GABA_AR clone 113-175; do not inhibit GABA_AR-mediated current; Kreye et al., JCI, 2021) produce similar effect that the ones presented in the initial study: a 24h incubation with each of these autoantibodies reduce the immunodetection of NMDAR and GABA_AR clusters. These data have now been added into the revised manuscript (Figure EV2).

2. *Expression of D1 seems minimal if not absent in CA1/CA2/CA3 pyramidal neurons (but present in dentate gyrus) based on several reports. Could the related D5 receptor mediate some of the effects ascribed to ectopically expressed D1 receptor?*

Response: The exact expression distribution of D1R in the hippocampus is still a matter of debate. Indeed, genetically-modified mice that express a fluorescent report with D1R or D1R promotor have shown very different outcomes, precluding any firm and valid conclusion. Importantly for our work, all studies convergence toward the fact that D1R are greatly expressed in the CA areas of the ventral hippocampus when compared to the dorsal one. In our study, we used cultured hippocampal neurons from dorsal and ventral hippocampi, indicating that most of our neurons likely express D1R. Furthermore, pharmacological approaches have consistently reported D1R/D5R or D1R only mediated effects in CA1/CA3 synapses. These studies consistently indicate a D1R-mediated transmission or regulation in these neuronal networks. To address the concern of the reviewer on this debate, we have added a sentence to comment on the heterogeneous expression of D1R (page 9) as well as on the putative role of D5R in the reported effect on the NMDAR-D1R interaction.

Dear Dr. Groc

Thank you for the submission of your revised manuscript to EMBO reports. It has been seen by former referee #3 who is very positive about the study and supports publication.

Browsing through the manuscript myself, I noticed a few editorial things that we need before we can proceed with the official acceptance of your study.

- Please update the 'Conflict of interest' paragraph to our new 'Disclosure and competing interests statement'. For more information see

<https://www.embopress.org/page/journal/14693178/authorguide#conflictsofinterest>

- Please remove the Author Contributions from the manuscript file and make sure that the author contributions in our online submission system are correct and up-to-date. The information you specified in the system will be automatically retrieved and typeset into the article. You can enter additional information in the free text box provided, if you wish.

- Reference format: et al needs to be used after 10 author names

- Heckmann & Geis 2023 is a reference to a preprint and should be labeled as such. The in text citation looks like this (preprint: Heckmann et al 2023) and in the reference list you need to add [PREPRINT] at the end of the citation. The citation also needs a doi.

(On a different note: I could not find the in-text citation. Is it part of a table?)

- Please write the abstract in present tense.

- Author checklist: please complete column D (select response: Yes or N/A).

- You describe the isolation of B cells from CSF of patients with autoimmune encephalitis or from a patient with NMDAR encephalitis in the methods section. Has informed consent been obtained from the patients and the procedure been approved? Please provide this information in the manuscript. Or has this information been given in the publications you cite (Kreye et al, 2016, 2021)? Please clarify.

- Experiments with Sprague-Dawley rats: please provide the details of the authority granting ethics approval and provide the reference number for approval.

- Data availability section: this section should exclusively refer to primary datasets deposited in public repositories. Therefore, please remove the statement 'The data supporting the findings are available within the article and its Supplementary Materials and are available from the corresponding author upon request. Reagent details are provided in the method section.' Please keep only the reference and linkout to the dataset on Bioline Archive ('The datasets produced in this study...').

- The information on funding needs to be part of the Acknowledgment section. All funding given in the manuscript needs to be specified in the online submission system. The information in the submission system must match that in the manuscript text.

- You refer to Suppl figure 2I on page 22 but this figure has not been supplied.

- Could you please upload the Reagents and Tools Table separately? The file type is 'Reagent Table' in the system.

- Our production/data editors have asked you to clarify several points in the figure legends (see below). Please incorporate these changes in the manuscript and return the revised file with tracked changes with your final manuscript submission:

A) Please note that a separate 'Data Information' section is required in the legends of all the figures. [The prefix Data information: ... is applied to information at the end of the legend that applies to several panels, e.g. statistical significance.]

b) Please note that the legend of figure 1L, M should be correctly labeled as L, M or L-M.

c) Please note that in figures 1i, j, l, m there is a mismatch between the annotated p values in the figure legend and the annotated p values in the figure file that should be corrected.

d) Please note that the box plots need to be defined in terms of minima, maxima, centre, bounds of box and whiskers, and percentile in the legends of figures 1j, m.

e) Please note that information related to n is missing in the legend of figure EV1b.

- Finally, EMBO Reports papers are accompanied online by A) a short (1-2 sentences) summary of the findings and their significance, B) 2-3 bullet points highlighting key results and C) a synopsis image that is 550x300-600 pixels large (width x height) in PNG or JPG format. You can either show a model or key data in the synopsis image. Please note that the size is

rather small and that text needs to be readable at the final size. Please send us this information along with the revised manuscript.

- On a different note, I would like to alert you that EMBO Press offers a new format for a video-synopsis of work published with us, which essentially is a short, author-generated film explaining the core findings in hand drawings, and, as we believe, can be very useful to increase visibility of the work. This has proven to offer a nice opportunity for exposure i.p. for the first author(s) of the study. Please see the following link for representative examples and their integration into the article web page:

<https://www.embopress.org/doi/full/10.15252/emj.2019103932>

With kind regards,

Referee #3:

The authors addressed the concerns well

The authors have addressed all minor editorial requests.

Dr. Laurent Groc
Interdisciplinary Institute for Neuroscience
CNRS - Universite de Bordeaux
146 rue Leo Saignat
Bordeaux, France 33077
France

Dear Laurent,

I am very pleased to accept your manuscript for publication in the next available issue of EMBO reports. Thank you for your contribution to our journal.

Best wishes and a Happy New Year,

Martina
